# Astrocytes promote a protective immune response to brain *Toxoplasma gondii* infection via IL-33-ST2 signaling

Katherine M. Still[1], Samantha J. Batista[1], Carleigh A. O'Brien[1], Oyebola O. Oyesola[2,3], Simon P. Früh[2], Lauren M. Webb[3], Igor Smirnov[1], Michael A. Kovacs[1], Maureen N. Cowan[1], Nikolas W. Hayes[1], Jeremy A. Thompson[1], Elia D. Tait Wojno[3], Tajie H. Harris[1] *

**1** Center for Brain Immunology and Glia, Department of Neuroscience, University of Virginia, Charlottesville, Virginia, United States of America, **2** Baker Institute for Animal Health and Department of Microbiology and Immunology, Cornell University, Ithaca, New York, United States of America, **3** Department of Immunology, University of Washington, Seattle, Washington, United States of America

* tajieharris@virginia.edu

**Data Availability Statement:** All relevant data are within the manuscript and its Supporting Information files.

## Abstract

It is of great interest to understand how invading pathogens are sensed within the brain, a tissue with unique challenges to mounting an immune response. The eukaryotic parasite *Toxoplasma gondii* colonizes the brain of its hosts, and initiates robust immune cell recruitment, but little is known about pattern recognition of *T. gondii* within brain tissue. The host damage signal IL-33 is one protein that has been implicated in control of chronic *T. gondii* infection, but, like many other pattern recognition pathways, IL-33 can signal peripherally, and the specific impact of IL-33 signaling within the brain is unclear. Here, we show that IL-33 is expressed by oligodendrocytes and astrocytes during *T. gondii* infection, is released locally into the cerebrospinal fluid of *T. gondii*-infected animals, and is required for control of infection. IL-33 signaling promotes chemokine expression within brain tissue and is required for the recruitment and/or maintenance of blood-derived anti-parasitic immune cells, including proliferating, IFN-γ-expressing T cells and iNOS-expressing monocytes. Importantly, we find that the beneficial effects of IL-33 during chronic infection are not a result of signaling on infiltrating immune cells, but rather on radio-resistant responders, and specifically, astrocytes. Mice with IL-33 receptor-deficient astrocytes fail to mount an adequate adaptive immune response in the CNS to control parasite burden–demonstrating, genetically, that astrocytes can directly respond to IL-33 *in vivo*. Together, these results indicate a brain-specific mechanism by which IL-33 is released locally, and sensed locally, to engage the peripheral immune system in controlling a pathogen.

## Author summary

*Toxoplasma gondii* is a highly successful parasite, estimated to infect one-third of the world's human population and many warm-blooded vertebrates. *T. gondii* traffics to the brain of its hosts where it persists for their lifetime. Immune pressure is required to

**Funding:** This work was funded by National Institutes of Health grants R01NS091067, R56NS106028, and R01NS112516 to T. H. H.; T32GM008328 to K. M. S. and J. A. T.; T32AI007046 to S. J. B.; T32AI007496 to C.A.O., M.N.C., and M.A.K.; T32GM007267 to M.A.K.; and R01AI32708 and R01AI130379 grants to E.D.T.W., and a University of Virginia School of Medicine R&D grant to T. H. H. The funders had no role in study design, data collection and analysis, decision to publish, or preparation of the manuscript.

**Competing interests:** The authors have declared that no competing interests exist.

control *T. gondii* in brain tissue, as evidenced by destruction of brain tissue in *T. gondii-infected* immunosuppressed patients. But how *T. gondii* presence is sensed by brain cells to orchestrate immune responses is not well understood. Here, we show that a host protein, IL-33, typically sequestered within brain cells in the healthy state, is released as a damage signal during brain *T. gondii* infection and can induce local changes to the brain environment to recruit immune cells. We show that astrocytes, specifically, are capable of directly responding to IL-33, thus illustrating a local mechanism by which brain-resident cells are alerted to pathogen entry.

## Introduction

Recruitment of immune cells to the brain during infection is a highly orchestrated process, requiring concerted expression of a number of chemokines and adhesion factors at the blood-brain barrier [1]. But the cues which precede these factors are less well understood. In particular, in many cases, it is unclear if brain resident cells possess the machinery to detect the presence of pathogens to promote the recruitment of peripheral cells. Murine infection with the eukaryotic parasite *Toxoplasma gondii* (*T. gondii*) features continual recruitment of blood-derived immune cells to the brain and serves as an excellent model for better understanding immune responses at this site.

*T. gondii* is a globally relevant pathogen which infects most warm-blooded vertebrates, including one-third of the human population [2–4]. Upon initial exposure of hosts to *T. gondii* through contaminated food or water [5], an early stage of infection occurs, called the acute phase, during which *T. gondii* disseminates throughout peripheral tissues [6]. By two-weeks post-infection, parasite has been largely cleared or controlled in most tissues, but ultimately persists in the brain of its hosts for their lifetime [2,5–9]. Mortality from *T. gondii* infection is associated with an increased prevalence of replicating parasite in brain tissue, documented in immunosuppressed patients undergoing transplant surgeries [10], and in HIV-AIDS patients [11–13], highlighting the importance of the immune response in controlling *T. gondii*. Indeed, control of brain *T. gondii* infection requires a Th1-dominated immune response [2,9], whereby CD4+ and CD8+ T cells and the IFN-γ they produce are required for survival [14]. Macrophages also exhibit anti-parasitic effector mechanisms which are necessary to control the parasite [9,15–21].

It is not known, however, how the parasite is sensed in the brain to create an environment that promotes immune cell entry, stimulation, and maintenance. During the acute phase of infection in the periphery, dendritic cells and macrophages can sense either the parasite itself or host signals to initiate chemokine and cytokine expression which recruits and skews a strong Th1 immune response [9,22–26]. However, resident dendritic cells and peripheral immune cells do not exist in brain tissue under steady-state conditions [27,28] and it is unclear if *T.gondii*-specific molecular patterns are sensed in this tissue. During chronic *T. gondii* infection of the brain, necrotic lesions form, characterized by the presence of replicating parasite, loss of brain-resident cell markers, and infiltration of immune cells, suggestive of tissue damage and alarmin release [14,29–33]. We hypothesized that indirect sensing of *T. gondii* infection, via recognition of host cell damage caused by the parasite, is an important step in instructing the immune response to *T. gondii* in the brain. Here we focused on the nuclear alarmin, IL-33, as a candidate orchestrator of the immune response to *T. gondii*. IL-33 is highly expressed in brain tissue [34], and IL-33 signaling has been shown to be protective against

tissue pathology during chronic *T. gondii* infection, but the mechanism by which IL-33 signals, and the immune mechanisms underlying this protection, were not studied in detail [35].

IL-33 is categorized as an alarmin because it is known to amplify immune responses upon signaling through its receptor ST2, also known as *il1rl1*, without a requirement for secretion or cleavage. A role for IL-33 in stroke [36,37], neurodegeneration [38,39], EAE [40,41], and CNS infection [42–44] has been described, but mechanistic understanding of IL-33 signaling during brain disease is limited. It is unclear in many instances which cell type(s) is the relevant responder to IL-33 during disease in the brain. While brain-resident macrophages have been shown to express the IL-33 receptor at baseline [45], astrocytes have been shown to upregulate it during pathology [37,46]. In addition, IL-33 receptor is also expressed on immune cells which can infiltrate the brain from the blood during disease [47,48]. Here, we separate the relative contribution of astrocytes, microglia/macrophages, and infiltrating immune cells in response to IL-33 release during brain pathology, finding that IL-33 can signal strictly within the brain to promote protective immunity to *T. gondii*.

## Results

### IL-33 is released during *T. gondii* brain infection and is required to control parasite

To study *T. gondii* brain infection, we infected mice by intra-peritoneal injection with the avirulent *T. gondii* strain Me49 and waited four weeks post-infection for a natural, chronic brain infection to be established. At this time point, infection of most other tissues throughout the body, also known as acute infection, has been controlled [6,7]. For all experiments, brain tissue was harvested at four weeks post-infection unless otherwise specified.

Consistent with its role as a pre-stored alarmin, IL-33 is highly expressed in the brain at baseline [34], and only mildly increases with infection (S1A Fig). Although astrocytes are the major source of IL-33 during development, oligodendrocytes become a significant contributor by postnatal day 30 [34,45,46,49]. In the *T. gondii*-infected brain, we found nuclear IL-33 to be expressed predominantly by mature, CC1+ oligodendrocytes and also by astrocytes, the frequencies of which varied by brain region (Fig 1A and 1B, S1B Fig). We found IL-33 to be expressed almost exclusively by oligodendrocytes in white matter tracts, such as the corpus callosum, while IL-33 expression in gray matter, such as the cortex, was split more evenly between astrocytes and oligodendrocytes (Fig 1A and 1B). These data align with IL-33 expression in the uninfected mouse brain [46], indicating that the major sources of IL-33 in the brain do not change with *T. gondii* infection. Collectively, these results indicate that IL-33 is expressed by glia in the *T. gondii* infected mouse brain parenchyma. Importantly, we also detected IL-33 protein expression in astrocytes in healthy human brain tissue, as has been shown previously on the transcript level [50](S1C and S1D Fig).

Since IL-33 does not need to be cleaved to be active, it is classically thought of as an alarmin that is released by necrotic cell death [51,52]. During *T. gondii* infection, the parasite itself as well as the inflammatory environment provide opportunity for host cell damage and alarmin release. During chronic infection, *T. gondii* predominantly exists in the brain as an intracellular cyst form which is slow growing [6,8] (Fig 1C) and does not appear to pose an immediate risk to cells, due to a lack of observed tissue destruction and inflammation surrounding cysts (Fig 1C). But, for reasons not fully understood, cysts can reactivate anywhere in the brain, releasing individual parasites previously contained within the cyst wall [8,30] (Fig 1C). Individual parasites can invade surrounding cells, replicate, and can lyse the cell or form a new cyst [53]. The presence of individual replicating parasites is correlated with morbidity in humans, and is thought to cause necrotic lesions in immunocompromised patients [10–13].

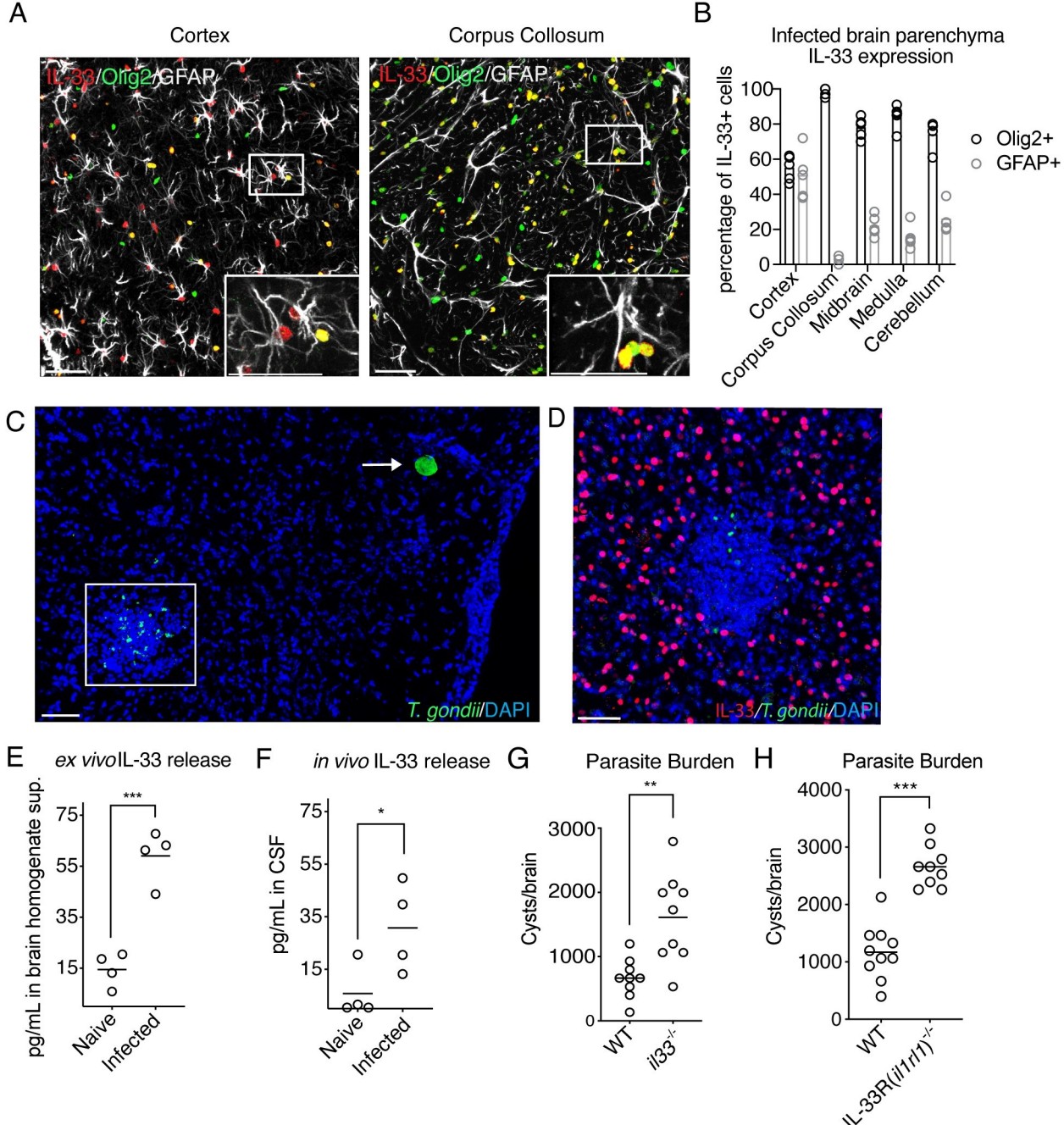

**Fig 1. IL-33 is expressed by oligodendrocytes and astrocytes during brain *T. gondii* infection, is released during infection, and is required for control of parasite. (A and B)** Representatitive images (A) and percentage quantification (B) of nuclear IL-33 protein expression (red) by cell type four weeks post-infection in various brain regions by confocal fluorescent microscopy. IL-33 is costained with the nuclear oligodendrocyte marker, Olig2 (green) or activated astrocyte marker, GFAP, (white). **(C and D)** Representative images of *T. gondii* in brain tissue, either in cyst form (arrow), or reactivated individual parasites (box)(C). IL-33-positive cells are absent from inflammatory foci containing replicating parasites (D). **(E and F)** Extracellular IL-33 release as measured by ELISA after *ex vivo* incubation of all cells isolated from infected brain tissue (E), or from CSF samples, each dot represents pooled CSF from 4–5 individual mice from a separate infection, displaying four infections in total (F). **(G and H)** Parasite burden as measured by cyst count from brain homogenate of IL-33-deficient (G) and IL-33 receptor (*il1rl1*)-deficient mice (H). Statistical significance was determined by two-tailed t-test (E) or randomized block ANOVA, when results from multiple independent experiments are shown (F-H). * = p < .05, ** = p < .01, *** = p < .001. Scale bars indicate 50μm.

Interestingly, we found clusters of immune cells, including T cells and macrophages, surrounding individual replicating parasites but not *T. gondii* cysts (S2A and S2B Fig, Fig 1C). Therefore, we hypothesized that either lytic *T. gondii* replication [53] or local inflammation, can cause release of local damage signals, such as IL-33. Corroborating this hypothesis, while IL-33 tiles evenly throughout uninfected brain tissue [46], we noted focal loss of IL-33 staining, as well as oligodendrocyte and astrocyte markers, at the center of inflammatory lesions containing parasite (Fig 1D, S2C and S2D Fig).

To assess if host cell death occurs in the *T. gondii* infected brain which could facilitate IL-33 release, we injected mice intraperitoneally with propidium iodide (PI), which is taken up by cells with loss of membrane integrity. 24 hours post-injection, while we did not observe significant clusters of PI positivity in uninfected (naïve) brain tissue (S2E Fig), we found focal areas of PI positivity in chronically infected brain tissue near inflammatory foci, denoted by a lack of GFAP staining and DAPI clustering (S2F Fig). These results indicate that infection of the brain by *T. gondii* can cause damage to host cells.

To determine if IL-33 was released during brain *T. gondii* infection, we first utilized an *ex-vivo* assay to measure extracellular IL-33 during infection. We processed naïve and infected mouse brains down to a single cell suspension containing brain resident cells as well as immune cells and parasite. We then incubated the single cell suspensions for four hours at 37˚C before taking the supernatant and measuring extracellular IL-33 by ELISA. Strikingly, IL-33 was present in detectable quantities from supernatants of infected brain samples, but not naïve controls, suggesting that *T. gondii* infection has the capacity to induce IL-33 release (Fig 1E). We then validated IL-33 release *in vivo*, by sampling cerebrospinal fluid (CSF) from the cisterna magna of infected mice and found detectable IL-33 levels in pooled CSF samples (Fig 1F).

We next asked if IL-33 was required to control infection. We infected wildtype and *il33*$^{-/-}$ mice and assessed their brains for parasite burden, enumerating a significantly increased number of cysts in *il33*-deficient mice (Fig 1G). We also detected an increased parasite burden in the brains of infected mice that lacked the IL-33 receptor (known as ST2 or *il1rl1*), by cyst count and quantitative PCR (Fig 1H, S3B Fig), demonstrating a role for extracellular IL-33 signaling during this infection. At 10 days post-infection, no parasite was detected in the peritoneal cavity of infected *il1rl1*$^{-/-}$ or wildtype animals. Therefore, *il1rl1*$^{-/-}$ mice are not delayed or defective in clearing parasite in the periphery. Additionally, parasite levels were equivalent in the brain at day 12 post infection (12DPI) in *il1rl1*$^{-/-}$ mice in comparison to wildtype mice (S3A Fig), further supporting that the control of parasites is intact during the early stages of infection. We did not find *il33*$^{-/-}$ or *il1rl1*$^{-/-}$ mice to succumb to infection. Nonetheless, these results demonstrate that extracellular IL-33 signaling plays a critical role in limiting *T. gondii* during chronic brain infection.

## IL-33 signaling is required for adequate numbers of functional T cells during chronic *T. gondii* infection

In order to better understand how IL-33 signaling was protective during *T. gondii* infection, we focused on characterizing *il1rl1*$^{-/-}$ mice from this point onward. We first profiled recruited immune cell populations in the brain, beginning with the adaptive immune response, because a strong, Th1-biased immune response is absolutely critical for control of chronic *T. gondii* infection [14]. In the absence of IL-33 signaling during infection, approximately one-third fewer total T cells were present in the brain by flow cytometry (Fig 2A and 2B), suggesting an inability to recruit or maintain these cells. Approximately 96% of T cells in *T. gondii* infected brains were positive for either CD4 or CD8 markers (Fig 2A), and there was no selective

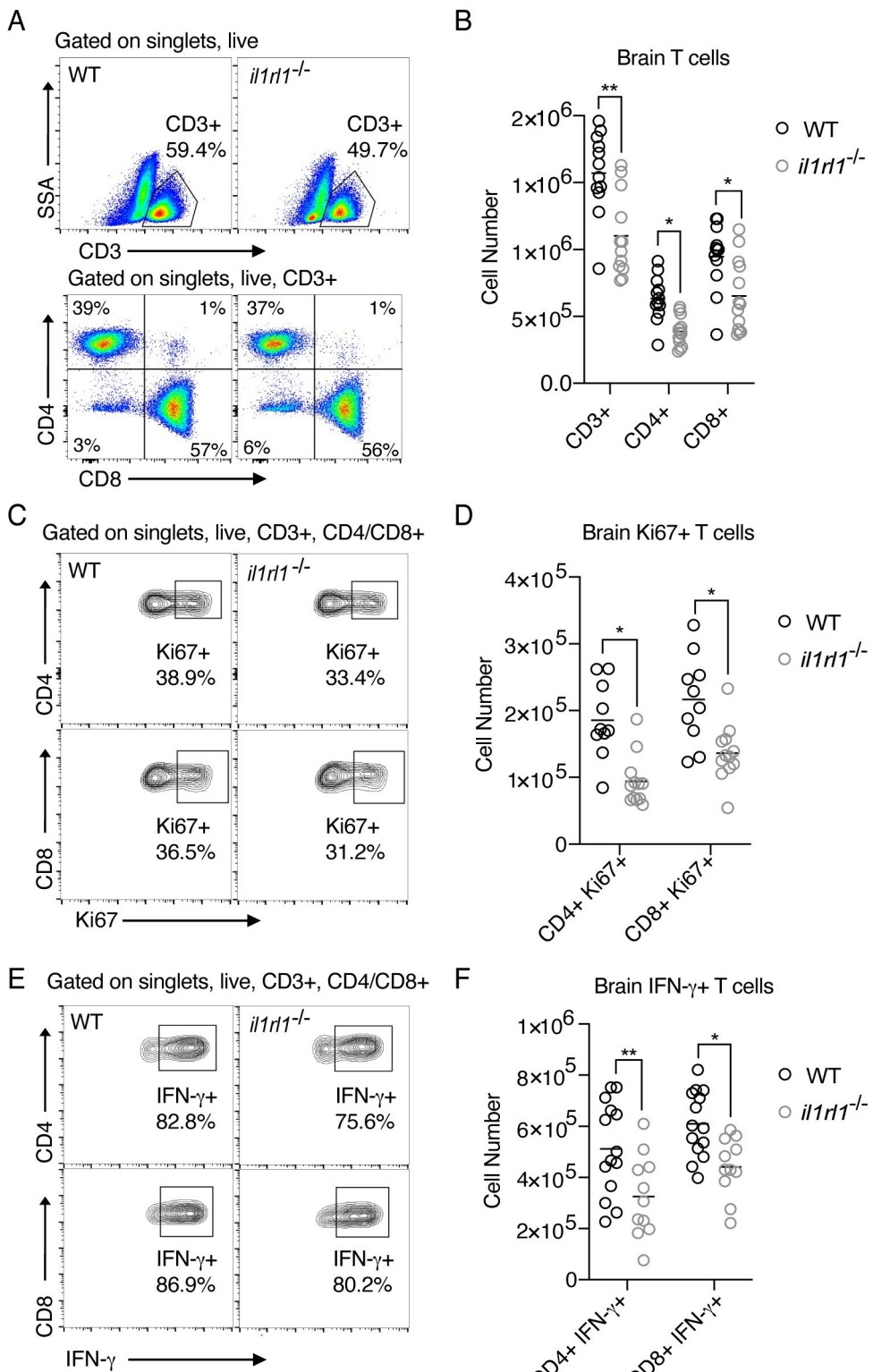

**Fig 2. IL-33 signaling is required for adequate numbers and functionality of T cells in the brain during chronic *T. gondii* infection. (A and B)** Representative flow cytometry plots (A) and quantification (B) of infiltrated total T cells in infected brain tissue four weeks-post infection. **(C and D)** Representative flow cytometry plots (C) and quantification (D) showing frequency of proliferating (Ki67+) T subsets in infected brain tissue. **(E and F)** Representative flow cytometry plots (E) and number (F) of IFN-γ-positive T cell subsets in infected brain tissue. IFN-γ was measured

following *ex-vivo* re-stimulation of brain cells, incubated with Brefeldin A and PMA/ionomycin for 5 hours at 37˚C (F). Statistical significance was determined by randomized block ANOVA (B,D,F), and each quantified panel displays three pooled independent experiments. * = p < .05, ** = p < .01, *** = p < .001.

decrease in either of these subsets, but rather a general reduction in T cell counts in *il1rl1*<sup>-/-</sup> mice (Fig 2A and 2B). A reduction in T cell numbers could be due to reduced recruitment, reduced survival, or reduced proliferation in the absence of IL-33 signaling. Indeed, fewer CD4+ and CD8+ T cells were proliferating in the brain (Fig 2C and 2D), and fewer of these cells were functional, displaying reduced production of the critical cytokine IFN-γ by *ex vivo* re-stimulation with PMA/ionomycin (Fig 2E and 2F). IFN-γ is critical because it induces widespread anti-parasitic changes, such as increasing chemokine production [54,55], adhesion factor expression [56], and intracellular killing mechanisms of infected cell types, such as macrophages [9]. We found that the chief sources of IFN-γ in the brain during infection were CD4+ and CD8+ T cells, with minimal contribution from NK cells, an important source of the cytokine during early stages of acute infection [57] (S4A Fig). These results show that extracellular IL-33 signaling is required to maintain an adequate anti-parasitic T cell response in brain tissue.

T cell numbers and function were unaffected at baseline in spleens of *il1rl1*<sup>-/-</sup> mice, in the spleen and the blood during chronic infection, as well as at sites of inflammation such as the peritoneal cavity during acute infection (S4B–S4E Fig). We did, however, note a decrease in T cell numbers in the blood at day 10 post-infection in *il1rl1*<sup>-/-</sup> mice (S4F Fig), and decreased serum IFN-γ at day 12 post-infection despite intact IL-12p40 levels (S4H and S4I Fig). These results indicate that IL-33 signaling does affect the T cell response; however, it does not appear that these defects underlie defects in brain immunity to *T. gondii*, since activated T cell numbers and parasite burden were unaffected in the brain at day 12 in *il1rl1*<sup>-/-</sup> mice (S3A and S6C Figs). Plasma IFN-γ levels at day 7 were also intact in *il1rl1*<sup>-/-</sup> mice (S4G Fig), and are likely enough to control parasite early in infection. Furthermore, a time course showed progressive decreases in IFN-γ+ T cell numbers in *il1rl1*<sup>-/-</sup> mice in comparison to wildtype mice (S6 Fig, Fig 2F), demonstrating brain-specific impacts on immunity in the absence of IL-33 signaling.

It is important to mention that decreased immune activation in *il1rl1*<sup>-/-</sup> mice, such as decreased IFN-γ production, during the acute or chronic stage of infection, could be beneficial in protecting against immunopathology. Indeed, *il1rl1*<sup>-/-</sup> mice are protected from weight loss, early mortality, and intestinal pathology during the acute stage of infection when mice are infected orally with *T. gondii* [58]. Infecting with 10 cysts of Me49 parasite intraperitoneally, we do not observe significant mortality or pathology in wildtype mice, but nonetheless, IL-33 signaling could potentiate immunopathology during more severe infections.

## IL-33 signaling is required for the recruitment and anti-parasitic function of myeloid cells during brain *T. gondii* infection

We were also interested in the impact of IL-33 signaling on the myeloid cell lineage since macrophages cluster tightly around replicating parasite in brain tissue (S2B Fig) and express iNOS, a key molecule involved in the control of *T. gondii* [9]. We assessed numbers of CD11b+ myeloid cells in infected *il1rl1*<sup>-/-</sup> mice by flow cytometry, using CD45hi expression to differentiate infiltrating myeloid cells from CD45int microglia (Fig 3A and 3B). Many of the CD45hi CD11b+ cells recruited to the *T. gondii*-infected brain are Ly6C+ CD11c- and Ly6G-, indicating that a large portion of recruited myeloid cells are Ly6Chi monocytes and Ly6Clo monocyte-derived macrophages [59]. Importantly, infected *il1rl1*<sup>-/-</sup> mice displayed a reduced frequency and number of CD45hi myeloid cells, by approximately half, in the brain during chronic infection, while CD45int cell numbers held constant (Fig 3A and 3B).

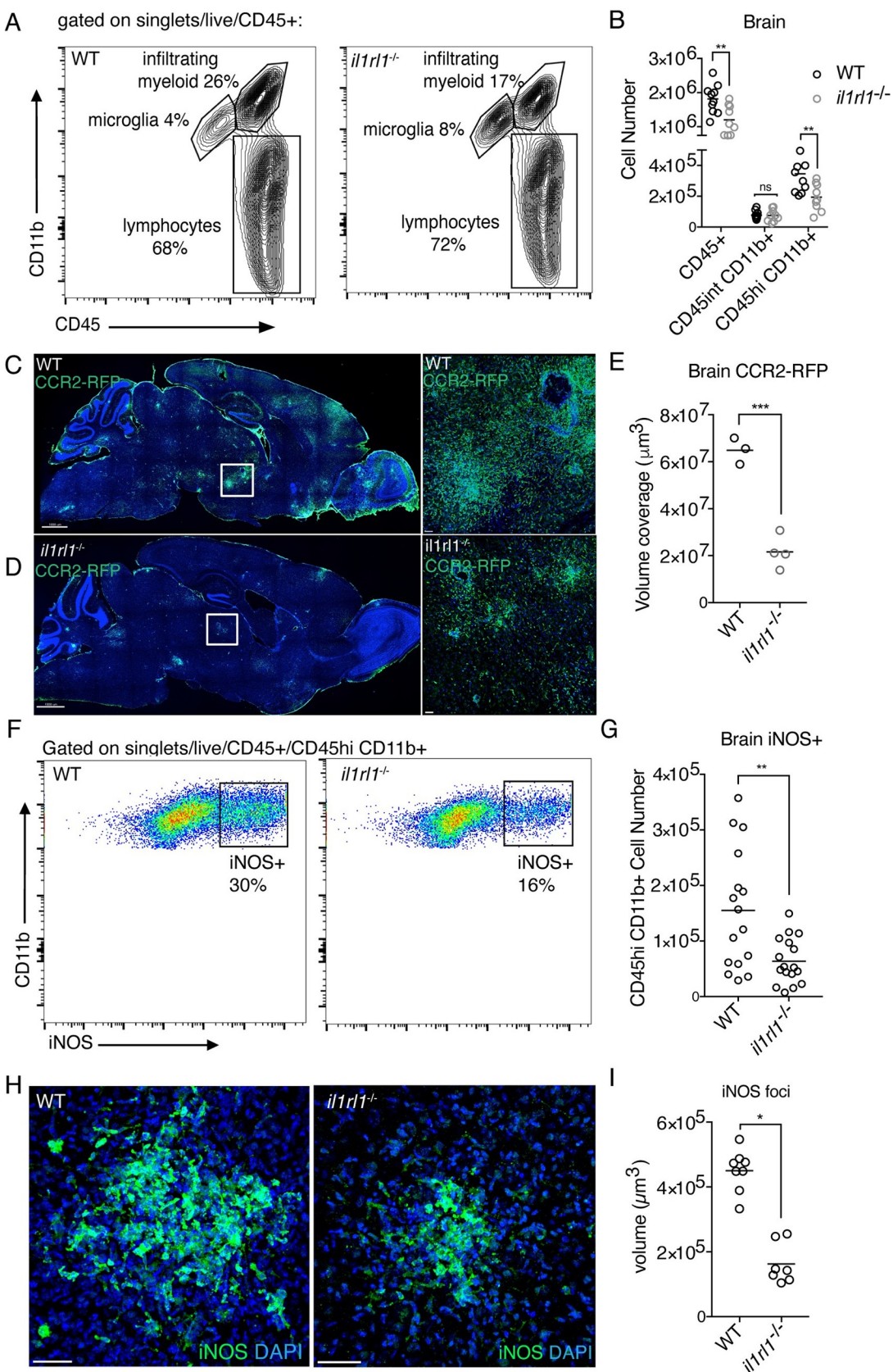

**Fig 3. IL-33 signaling is required for the recruitment and anti-parasitic function of peripheral myeloid cells in the brain during chronic *T. gondi* infection. (A and B)** Representative flow cytometry plots (A) and quantification (B) of CD11b + myeloid cells in the brain 4 weeks post infection. CD45hi expression was used to differentiate infiltrating myeloid cells from CD45int microglia. **(C-E)** Visualization (C and D), and quantification (E) of infiltrated CCR2+ monocytes by fluorescence confocal microscopy of infected CCR2-RFP reporter mice. *il1rl1*-deficient mice were crossed to CCR2-RFP mice to assess the contribution of IL-33-*il1rl1* signaling to monocyte recruitment. **(F and G)** Representative flow cytometry plots (F) and quantification (G) of iNOS+ CD45hi, CD11b+ infiltrating cells in the brain. **(H and I)** Visualization (H) and quantification (I) of the size of iNOS foci in brain tissue by fluorescence confocal microscopy. Statistical significance was determined by randomized block ANOVA when two or more experiments were pooled (B, G, I), and by a two-tailed t-test (E). * = p < .05, ** = p < .01, *** = p < .001. Scale bars indicate 2000μm (C,D) and 50μm (H).

To more specifically assess monocyte recruitment, we used CCR2-RFP reporter mice to visualize monocytes in the brain during chronic infection. Monocyte-derived cells are critical for control of brain *T. gondii* infection, as anti-CCR2 antibody administered during chronic infection results in rapid mortality [59]. By immunohistochemistry, we observed large numbers of CCR2+ cells throughout the brain 4 weeks post-infection in wildtype mice (Fig 3C). Importantly, we crossed CCR2-reporters to a *il1rl1*-/- background, which revealed a marked reduction in CCR2+ cells (Fig 3D and 3E). These results confirmed that there is a defect in monocyte recruitment in the absence of IL-33 signaling during infection.

Next, we assessed the anti-parasitic function of the myeloid compartment in the absence of IL-33 signaling, by focusing on inducible nitric oxide synthase (iNOS) expression. Synthesis of nitric oxide by host cells can deprive the parasite of essential amino acids and prevent parasite growth *in vitro* [60]. *In vivo*, iNOS knockout mice succumb to infection during the early chronic phase [16]. Of the myeloid cells that were able to infiltrate the brain, fewer of these cells were making iNOS in the absence of IL-33 signaling (Fig 3F–3I). Interestingly, we did not detect significant iNOS positivity in any tissues, aside from the brain, in acute or chronic infection (S5 Fig). This finding is supported by literature demonstrating that iNOS is particularly important during the chronic phase, but not the acute phase of *T. gondii* infection [16], for reasons that are unclear. No defects in myeloid cell number were found at baseline in uninfected *il1rl1*-/- mice, in peripheral tissues during acute infection, including in the peritoneum, blood, and spleen, or in peripheral tissues during chronic infection (S5 Fig). In sum, these results, in conjunction with T cell deficits, demonstrate that IL-33 signaling impacts the presence and function of immune cell populations that are necessary for controlling *T. gondii* infection in the brain.

Finally, because we noted defects in both IFN-γ+ T cells and iNOS+ macrophages in *il1rl1*-/- mice, we wanted to better understand the kinetics of the reliance of immune cell populations on IL-33 signaling. Thus, we performed a time course and assessed T cell and myeloid cell numbers and activation in the brain at various stages of infection. We found that at day 12 post-infection, when significant immune populations are first present in the brain, immunity was largely intact in *il1rl1*-/- mice, measured by IFN-γ+ T cells and INOS+ myeloid cells (S6A–S6E Fig). By day 21, IFN-γ+ T cell numbers were significantly reduced in *il1rl1*-/- mice in comparison to wildtype mice, and infiltrating myeloid cell numbers were reduced as well (S6F–S6I Fig). However, at day 21 post-infection, iNOS frequencies were intact in *il1rl1*-/- mice (S6J Fig), in contrast to a significant decrease of iNOS frequency in il1rl1-/- mice at day 28 post infection (Fig 3F and 3G). These results indicate that iNOS expression is one of the last phenotypes to be affected by IL-33 signaling.

## IL-33 signaling induces expression of factors involved in recruitment of immune cells to the brain

Next, we asked if IL-33 signaling was changing the environment within the brain to make the tissue conducive to immune cell recruitment. Initiation and maintenance of an immune

response in the brain requires expression of cytokines, chemokines, adhesion factors, and factors which promote the entry of and maintain proliferation of immune cells [1]. It is well known that *T. gondii* infection induces many of these factors [54–56,61],which we validated by whole brain qRT-PCR. We find a profound increase in brain *ccl2*, *cxcl9*, *cxcl10*, *ccl5*, *cxcl1*, *vcam*, and *icam* expression over uninfected controls (Fig 4A). When we assessed the same genes in infected *il1rl1*-deficient mice, we saw significantly decreased expression of the chemokines *ccl2*, *cxcl10*, and *cxcl1* (Fig 4B), along with smaller reductions in the expression of the adhesion factors *vcam* and *icam* (Fig 4B). Interestingly, *ccl2* and *cxcl10* expression has been attributed to astrocytes by *in situ* hybridization during *T. gondii* infection [55]. Although *cxcl1* has not been extensively studied in chronic *T. gondii* infection, it has also been reported in astrocytes during neuroinflammation [62]. However, we did not observe an effect of IL-33 signaling on *cxcl9* expression, a chemokine which is made by PU.1-expressing cells rather than astrocytes [63] (Fig 4B). These results suggest that IL-33 signaling, either directly or indirectly, could be inducing chemokine expression in astrocytes during *T. gondii* infection.

We next wanted to identify the cellular source of chemokine in infected brain tissue to better understand where IL-33 was exerting its effects. We focused on studying the expression pattern of a chemokine whose transcript levels were altered to the greatest degree in the absence of IL-33 signaling, the monocyte chemoattractant *ccl2*. We used immunofluorescence microscopy to image the brain tissue of chronically infected *ccl2*-mCherry reporter mice [64]. We observed *ccl2*-mCherry expression in "hotspots" throughout the brain, implicating a local response to signals in brain tissue (Fig 4C). We validated by immunohistochemistry that *ccl2* expression was greatly reduced in *il1rl1*$^{-/-}$ mice by crossing our *ccl2*-mCherry reporters to an *il1rl1*-deficient background (Fig 4C–4E). Specifically, *ccl2* foci in *il1rl1*$^{-/-}$ mice were much reduced in size compared with wildtype infected mice (Fig 4D and 4E), while *ccl2* expression was unchanged between groups at baseline in uninfected mice (S7A Fig). Of note, we also observed astrocytes to co-express IL-33 and *ccl2*, but only approximately 25% of *ccl2*+ astrocytes in the infected cortex expressed IL-33 (S7B and S7C Fig). Although multiple signals undoubtedly converge to induce chemokine expression during infection, including other innate cytokines [65], these data suggest that IL-33 is a major contributor to the induction of local *ccl2*, supporting a role for IL-33 in inducing chemokine production to promote immune cell entry to the brain.

To get a better understanding of which cells might be responding directly or indirectly to IL-33 to induce *ccl2* expression, we next assessed which cells were producing the *ccl2*, and found that *ccl2*-mCherry in infected brain tissue co-localized with both GFAP+ astrocytes and Iba1+ macrophages (Fig 4F). Approximately 75% of cells expressing *ccl2* were astrocytes, 22% were Iba1+ macrophages, and 3% of cells did not co-stain with either of these markers (Fig 4G). These results were validated by assessing *ccl2* expression in brain cells magnetically enriched for either astrocytes or macrophages. To enrich for astrocytes, we used a previously validated protocol, which first removes CD11b+ cells by negative magnetic isolation, followed by positive isolation of ACSA-2 positive cells [66](S8A–S8C Fig). At the same time, we enriched for CD11b+ myeloid cells by keeping the CD11b+ magnetically-enriched fraction (S8A–S8C Fig). RNA from these cell populations showed a 10-fold higher *ccl2* expression in astrocytes compared with macrophages during chronic infection (Fig 4H). Thus, because *ccl2* expression is dependent on IL-33 signaling, and because astrocytes are the major producer of *ccl2* in the infected brain, we hypothesized that IL-33 can signal directly on astrocytes to induce chemokine expression, or signals through an intermediate cell type to impact astrocytic chemokine expression.

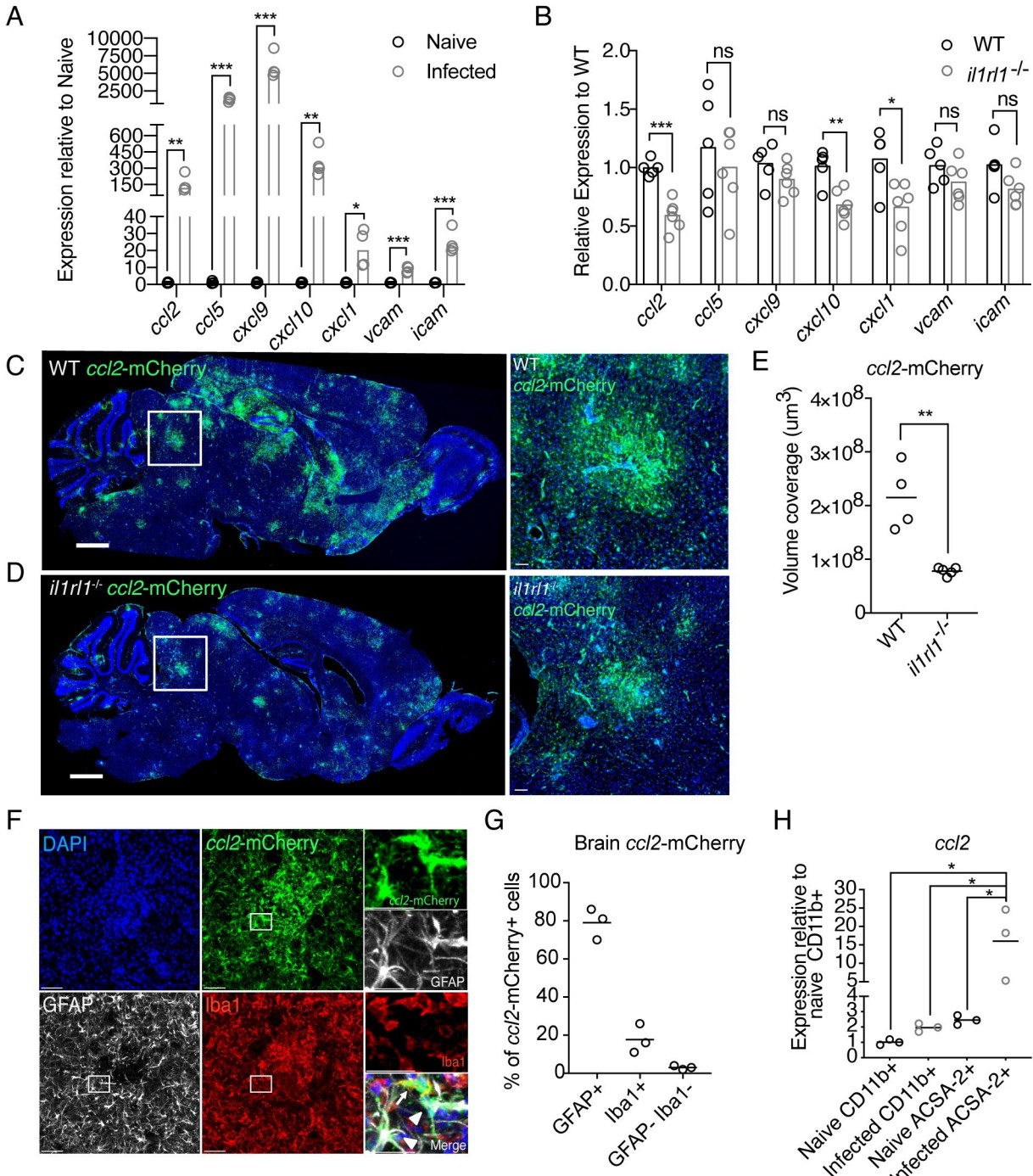

**Fig 4. IL-33 signaling induces factors which recruit immune cells to the brain. (A and B)** Real time PCR from whole brain homogenate was used to assess changes in chemokine and adhesion factor gene expression from naïve to infected animals (A) and between wildtype and *il1rl1*-deficient animals (B) at four weeks post infection. **(C-E)** Visualization (C,D) and quantification (E) of *ccl2*-RFP reporter expression by mCherry staining (green) in infected brain tissue by confocal fluorescence microscopy. **(F and G)** Breakdown of *ccl2*-mCherry reporter positivity by cell type using confocal microscopy. mCherry (green) is colocalized with Iba1+ macrophages (red) or activated, GFAP+ astrocytes (white). **(H)** Cell-type specific magnetic enrichment for myeloid cells (CD11b+) or astrocytes (CD11b- and ACSA-2+) in naïve and chronically infected brain tissue. Single cell suspensions of enriched cells were resuspended in Trizol, RNA extracted, and run by real time PCR for *ccl2* expression. Statistical significance was determined by two-tailed t-test (A, B, E) or One-way ANOVA with Tukey's test (H). * = p < .05, ** = p < .01, *** = p < .001. Scale bars indicate 2000μm (C,D) and 50μm (F).

## IL-33 signals on a radio-resistant cell type to control chronic *T. gondii* infection

In order to further understand IL-33 signaling in the brain, we began to narrow down the cell type(s) that directly responds to IL-33 to support protective immune responses. In studies concerning IL-33 during CNS disease, genetic evidence for a responding cell type to IL-33 is lacking. This is likely due in part to the fact that several brain-resident cells can express ST2 (*il1rl1)* during disease, such as microglia, astrocytes, and endothelial cells [37,67–69], but a wide range of immune cells can also express the receptor, including but not limited to type 2 innate lymphoid cells (ILC2s), regulatory T cells, Th2 cells, mast cells, and macrophages [48,70,71]. During *T. gondii* infection, we did not detect ST2 expression on the immune cells present in highest numbers in the infected brain–effector T cells and monocyte-derived myeloid cells (S9 Fig). We did, however, detect ST2 expression on ILC2s, which decreased significantly in frequency from uninfected to infected brain tissue, as well as mast cells, and regulatory T cells (S9 Fig).

To determine if IL-33 signals on a radio-sensitive or a radio-resistant cell type to exert its effects during chronic *T. gondii* infection, we lethally irradiated wildtype and *il1rl1*$^{-/-}$ mice and reconstituted these mice with bone marrow from either wildtype donors or *il1rl1*$^{-/-}$ donors (Fig 5A). Blood was assessed for reconstitution prior to infection (Fig 5B–5D), and in all cases, immune cells were >90% positive for the congenic CD45 marker of donor mice, confirming successful reconstitution of the chimera (Fig 5B–5D). We note that irradiation did alter the severity of infection, resulting in high parasite burdens in all groups, and a high number of infiltrating cells (Fig 5E–5G) Nonetheless, these experiments revealed that *il1rl1*-deficiency on radio-resistant cells, or in il1rl1$^{-/-}$ recipient, but not donor mice, recapitulated the phenotype of global knockouts, including reduced infiltrating myeloid and T cell numbers, and parasite burden, whereas *il1rl1* expression on donor bone marrow, was dispensable (Fig 5E–5G). These data demonstrate that IL-33 signals predominantly on non-hematopoietic cells to control brain *T. gondii* infection, thus largely ruling out immune cells as responders.

Importantly, microglia are not fully replenished by bone marrow-derived cells post-irradiation. Therefore, we cannot rule out microglia as responding to IL-33 in the brain. In addition, microglia highly express *il1rl1* at baseline [37,45,46,67], prior to chronic infection. To better understand which radio-resistant cells were capable of responding to IL-33, we assessed *il1rl1* expression on microglia/macrophages, and ACSA-2+ astrocytes, which have also been shown to express the IL-33 receptor, especially during disease [37,46,67]. To do this, we magnetically enriched for these cell types from naïve and chronically infected brain tissue, and found that microglia/macrophages cells express *il1rl1* at high levels in uninfected mice, but downregulate the IL-33 receptor 20-fold upon infection, while astrocytes express low levels at baseline and increase receptor expression with infection (S10A Fig). This result indicates a change in capability of cells which are able to respond to IL-33 prior to and following infection. Other brain resident cells, such as endothelial cells, neurons, and oligodendrocytes may express the IL-33 receptor during *T. gondii* infection and contribute to immunity. We chose to focus on microglia/macrophages and astrocytes because they have been previously reported to be the predominant cells types that express the IL-33 receptor [37,45,46,49,67,68,72], are major producers of IL-33-dependent chemokine during *T. gondii* infection [55] (Fig 4G and 4H), and are required for control of *T. gondii* infection [73,74].

## IL-33 signaling on astrocytes, but not microglia/macrophages, potentiates immune responses and limits parasite burden

To determine if IL-33 directly signals on microglia/macrophages or astrocytes, we crossed *il1rl1*$^{fl/fl}$ mice [75], to either constitutive CX3CR1cre mice or GFAPcre mice, respectively. We

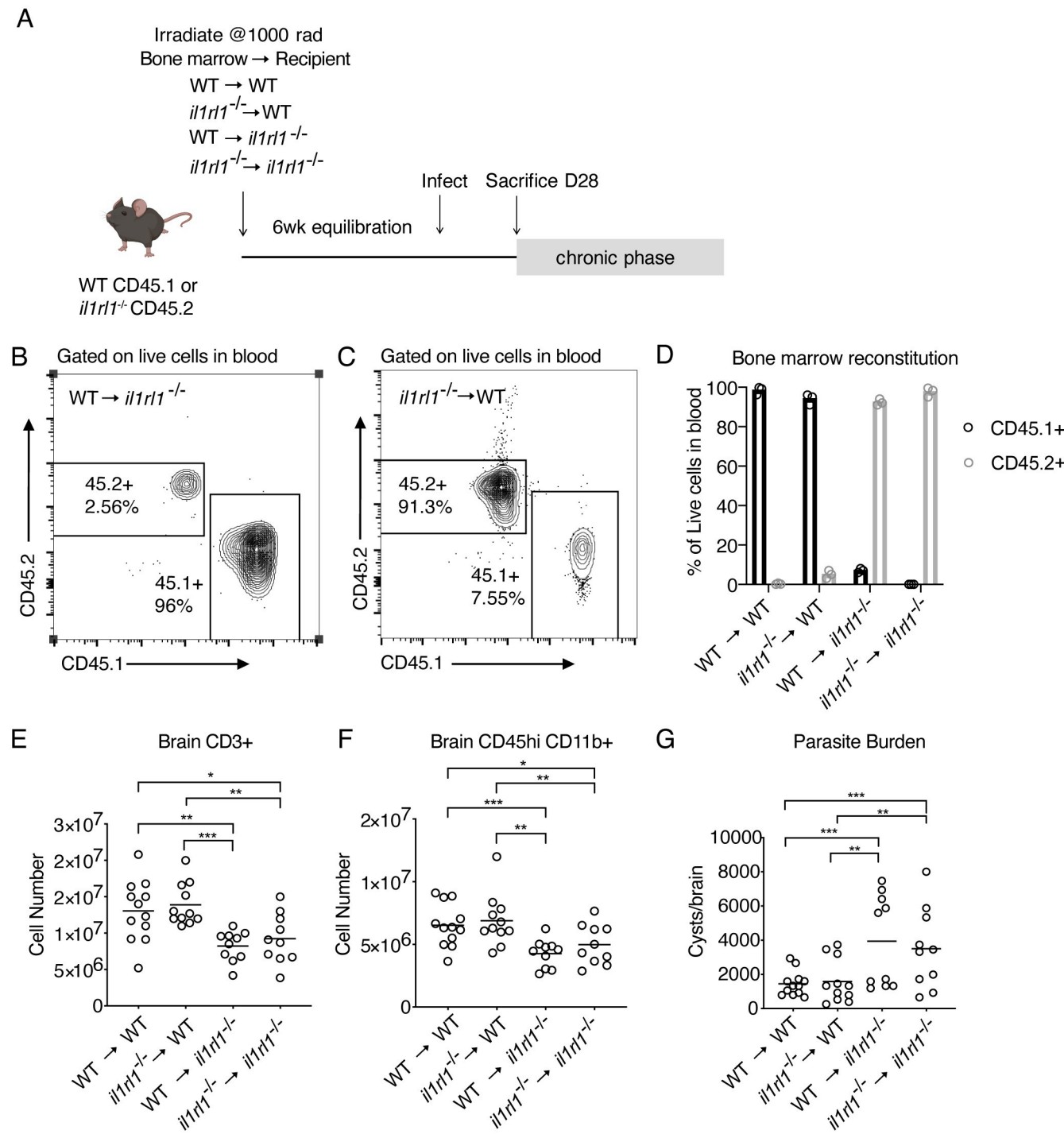

**Fig 5. IL-33 signals on a radio-resistant responder. (A)** Bone marrow chimera experimental setup. **(B-D)** Representative flow cytometry plots (B,C) and quantification (D) of bone marrow reconstitution in blood 6 weeks post irradiation and prior to infection. **(E and F)** Quantification of immune cells, including T cells (E) and infiltrating myeloid cells (F) in the brain at 4 weeks post infection following irradiation. **(G)** Parasite burden as assessed by cyst count of brain homogenate. Statistical significance was determined by randomized block ANOVA (E-G), each panel showing data pooled from two independent experiments * = p < .05, ** = p < .01, *** = p < .001.

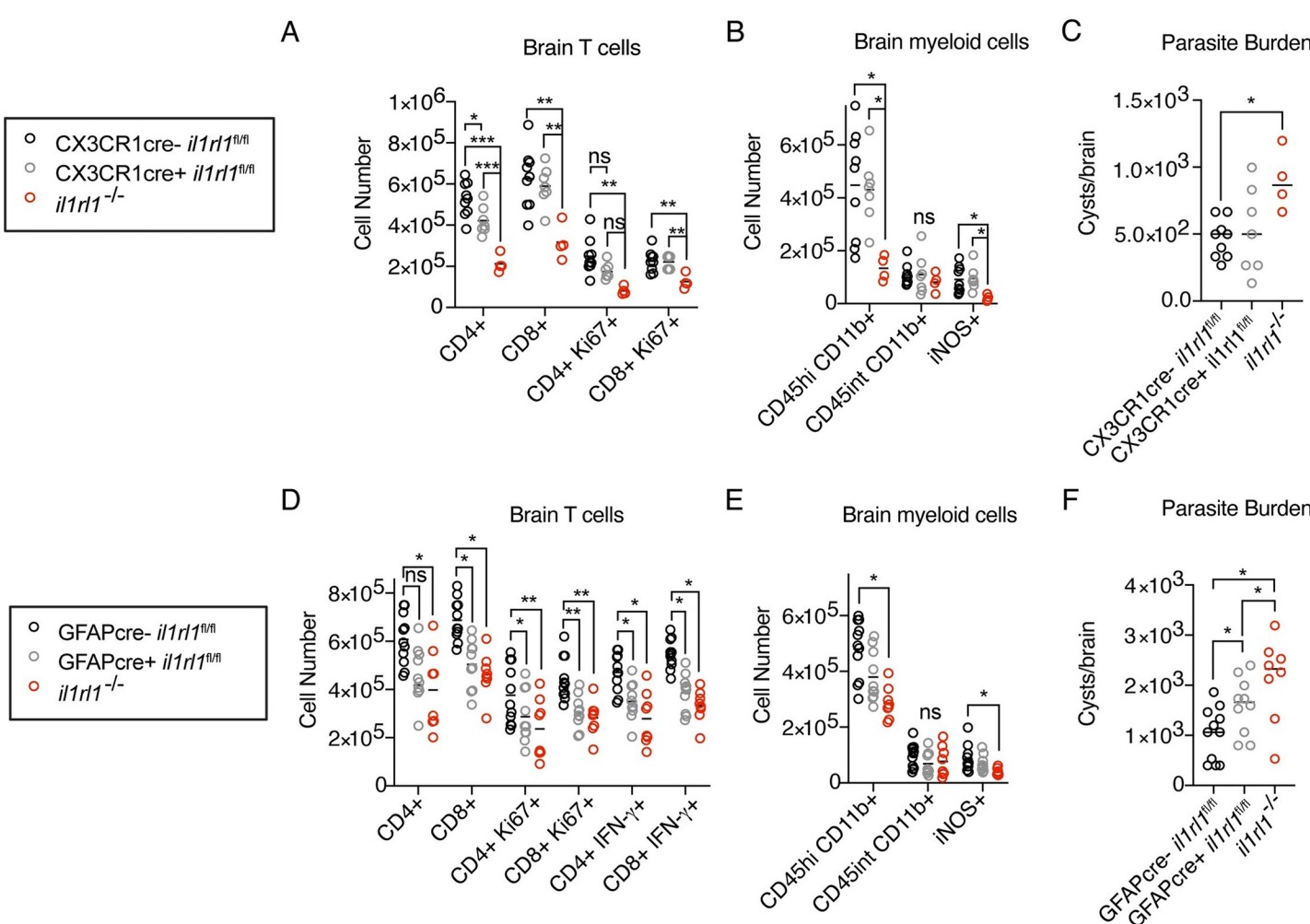

**Fig 6. IL-33 signaling on astrocytes, but not macrophages, is required to control brain *T. gondii* infection. (A and B)** Quantification of brain immune cells in CX3CR1cre *il1rl1*fl/fl mice by flow cytometry–including T cell subsets (A) and myeloid cells (B) at four weeks post infection. **(C)** Parasite burden in CX3CR1cre *il1rl1*fl/fl mice as assessed by cyst enumeration in brain homogenate. **(D and E)** Quantification of brain immune cells in GFAPcre *il1rl1*fl/fl mice by flow cytometry–including T cell subsets (D) and myeloid cells (E). IFN-γ was measured following *ex-vivo* re-stimulation of brain cells, incubated with Brefeldin A, and PMA/ionomycin for 5 hours at 37˚C. **(F)** Parasite burden in GFAPcre *il1rl1*fl/fl mice as assessed by cyst enumeration in brain homogenate. Statistical significance was determined by one-way ANOVA with Tukey's test (A-C), or by randomized block ANOVA(D-F), where each panel shows data pooled from two independent experiments * = p < .05, ** = p < .01, *** = p < .001.

observed that deletion of *il1rl1* from macrophages did not affect any major phenotype observed in *il1rl1*⁻/⁻ knockout mice, including brain myeloid cell number and function, T cell number, or parasite burden (Fig 6A–6C). We observed minor defects in CD4+ T cell number in these mice, including CD4+ T cell proliferation (Fig 6A). When we deleted *il1rl1* from astrocytes (S10B Fig), while myeloid cell recruitment and activation was relatively unaffected, the adaptive immune response was impacted, including reduced T cell proliferation and IFN-γ production, in mice with IL-33R-deficient astrocytes (Fig 6D and 6E). In contrast with CX3CR1cre mice, GFAPcre *il1rl1*fl/fl mice exhibited stronger CD8+ T cell proliferation deficits than CD4+ T cells (Fig 6D), perhaps suggesting an impact of IL-33 signaling on antigen presentation. Importantly, aligned with a decrease in T cell-derived IFN-γ, which is critical for control of infection [14], GFAPcre *il1rl1*fl/fl mice displayed increased parasite burden (Fig 6F). GFAPcre *il1rl1*fl/fl mice did not have defects in immune cell compartments in other tissues,

such as the spleen, displaying increased immune cell numbers in the spleen during chronic infection (S10C Fig), to a level comparable with whole-body *il1rl1*-deficiency. Collectively, these results show that astrocytes respond to a damage signal, IL-33, to promote a protective immune response to *T. gondii* in the brain.

## Discussion

We observe that brain resident cells can respond to IL-33, a damage-associated molecular pattern (DAMP), to initiate protective immunity during chronic *T. gondii* infection. This work begins to uncover the mechanisms by which the presence of *T. gondii* can be sensed specifically in brain tissue, which is devoid of peripheral immune cells in the healthy state. Indeed, much of the work on sensing of *T. gondii* has focused on immune cell recognition of *T. gondii* in peripheral tissues, during the acute phase of infection. It has been shown that *T. gondii* profilin protein, a pathogen-associated molecular pattern (PAMP), can be sensed directly by TLR11 and TLR12-expressing dendritic cells in the periphery, which are capable of initiating protective Th1 immunity by producing IL-12 [9,22,23]. In humans, however, TLR11 is a pseudogene and TLR12 is not expressed [76]. Thus, more recent studies have focused on innate sensing mechanisms with potential relevance to humans, including NOD-like receptor sensing of *T. gondii* [24,25], as well as alarmin recognition, such as S100A11 [26], by macrophages. Since we find IL-33 to be highly expressed in human brain tissue, we hypothesize that our results will be relevant to control of human brain infection. Furthermore, our results shed light on the importance of continual DAMP recognition to contain infection long after adaptive immune responses are first primed in peripheral tissues.

We found nuclear IL-33 to be expressed by oligodendrocytes and astrocytes in the mouse *T. gondii*-infected brain, and we were able to detect extracellular IL-33 in the cerebrospinal fluid during chronic infection. Since extracellular IL-33 is negatively regulated in a number of ways to limit inflammation [51,77,78], this result suggests high levels of IL-33 release during brain *T. gondii* infection. Our results also raise the question of how IL-33 is liberated from astrocytes and oligodendrocytes during disease. Although our results suggest that host cell death, indicated by propidium iodide staining, is occurring in the *T. gondii* infected brain which could release IL-33, the possibility of secretion of IL-33 from living cells cannot be ruled out [71]. Mechanisms of glial death during disease have not been extensively studied *in vivo*, and it is unclear if the parasite itself, or if secondary, toxic inflammation can mediate the release of host damage signals. It is also unclear how long-ranging the effects of alarmin signaling are. Work on IL-33 in cardiac muscle [75], as well as the retina [79], has indicated that IL-33 often signals very locally, acting in either an autocrine or paracrine manner. For example, Müller cells, specialized astrocytes in the retina, have been implicated in the release and sensing of IL-33 in the retina [79]. Our work further corroborates local IL-33 signaling, as IL-33-dependent *ccl2* expression is found adjacent to necrotic lesions containing replicating *T. gondii*. Thus, IL-33 may be released by astrocytes or oligodendrocytes, and can signal on astrocytes, which could be the same cell or an adjacent cell, to induce chemokine expression necessary for the recruitment of anti-parasitic immune cells to the brain.

In the lung, gut, and skin, IL-33 has historically been shown to signal almost exclusively on immune cells to orchestrate immunity [48]. However, recent work demonstrates that IL-33 can signal on non-hematopoietic cells–including cardiomyocytes in the heart [75] and endothelial cells in culture [69,80]. We found, consistent with a previous report in stroke [37], that IL-33 signals on a radio-resistant responder during brain *T. gondii* infection. Tissue microenvironment and context likely influence whether IL-33 signaling on immune cells is relevant. Peripheral immune cells are not present in healthy brain tissue due to the blood brain barrier,

thus necessitating the capability of brain resident cells to respond to damage. Additionally, the environmental milieu specific to CNS *T. gondii* infection promotes a robust Th-1skewed immune response necessary for intracellular parasite killing, characterized by T-cell derived IFN-γ and macrophage-derived iNOS [7,9]. Thus, the IL-33-dependent mechanisms, specifically those activating type 2 immune responses that drive asthma, allergy, and the expulsion of helminth infections [47,70], may not be relevant to brain infection with an intracellular pathogen.

When we examined brain-resident cell expression of *il1rl1*, we found that microglia express high levels of *il1rl1* in healthy brain tissue, but they downregulate the receptor 20-fold upon infection. A downshift in microglia inflammatory genes has been recently reported by our laboratory during *T.gondii* infection [73], and it is unclear what factors drive this process. However, these results do not rule out the importance of IL-33 signaling on microglia in other settings. Indeed, during development, microglia expression of *il1rl1* is required for microglial phagocytosis of synapses [45]. While we did not find any major impact of microglia/macrophage *il1rl1* expression on control of chronic *T. gondii* infection, it is possible that microglia respond to IL-33 at earlier time points, such as when the parasite first reaches the brain. If microglia normally respond to IL-33 earlier in infection, our results indicate that astrocyte responses to IL-33, or additional MyD88-dependent signals, may compensate for the lack of *il1rl1*-signaling in microglia.

In contrast to microglia/macrophages, we found that astrocytes increase expression of the IL-33 receptor upon infection. Astrocytes are critical for controlling CNS infection and potentiate inflammation. For example, astrocyte responses to IFN-γ are necessary to control *T. gondii* infection [74]. We found that IL-33 receptor expression on astrocytes was required for adequate promotion of an anti-parasitic immune response in the brain, including proliferation and IFN-γ production in T cells, which is critical for control of *T. gondii*. This is the first time, to our knowledge, that astrocytes have been directly implicated in an IL-33 response via genetic manipulation. Our results also raise questions regarding the role of astrocytes in supporting T cell responses to infection and whether these responses are direct or indirect. IL-33 signals through the adaptor MyD88 [70], and could influence the expression of numerous molecules that support T cell responses, including trophic factors, TCR engagement, chemokines, or blood brain barrier opening for T cell entry.

Notably, immune defects in GFAPcre *il1rl1*<sup>fl/fl</sup> mice did not match the magnitude observed in *il1rl1*<sup>-/-</sup> mice, and did not significantly impact myeloid cell recruitment. These results either implicate compensation by other cell types when *il1rl1* is deleted from a single population, or the presence of additional *il1rl1*-expressing radio-resistant cell types not considered here, such as neurons, oligodendrocytes, and endothelial cells, which have recently been shown to express *il1rl1* during disease and can be directly activated in response to IL-33 *in vitro* [69,80]. Additionally, it is possible that a decrease of circulating IFN-γ at day 12 post infection in *il1rl1*<sup>-/-</sup> mice has lasting impacts on chemokine and adhesion expression responsible for recruitment of myeloid cells. Nonetheless, our results indicate that astrocytes are capable of responding to extracellular IL-33 during infection. This finding, in addition to the detection of IL-33 release in the brain, has broad implications for sensing of tissue damage in other disease states, as definitive evidence that alarmins can signal strictly within brain tissue to initiate protective immunity.

Our results underscore the importance of the signaling of one alarmin in controlling chronic *T. gondii* infection. Little is known about the direct recognition of *T. gondii* in brain tissue and we implicate the response to infection-induced damage as a major mechanism of innate immune activation in the brain throughout chronic infection. Numerous canonical alarmins are expressed in the brain and their receptors are differentially expressed by cell type.

Thus, there is likely a complex integration of multiple PAMP and DAMP signals that is required to allow peripheral immune cells to overcome the blood brain barrier, infiltrate the tissue parenchyma, and carry out effector function. Identifying the cell types that can respond to specific DAMPs in the brain will be key not only to understanding immunity to CNS infection but any neuroinflammatory condition with hallmarks of cell death, including neurodegeneration.

## Methods

### Ethics Statement

Animal care and procedures adhered to the regulations of the Institutional Animal Care and Use Committee (IACUC) at the University of Virginia under protocol number 3968 and at Cornell University under protocol number 2014–0119. For human tissue samples, consent was not obtained due to anonymity of samples.

### Contact for reagent and resource sharing

Further information and requests for resources and reagents should be directed to and will be fulfilled by the Lead Contact, Tajie Harris (tajieharris@virginia.edu).

### Experimental mice

C57BL/6, CCL2-RFP^flox (Jackson Stock No: 16849), CCR2^RFP (Jackson Stock No: 017586), GFAPcre (Jackson Stock No. 024098), and CX3CR1cre mice (Jackson Stock No. 025524) were purchased from Jackson Laboratories. B6.SJL-Ptprc^a Pepc^b/BoyCrCl (C57BL/6 Ly5.1) mice were purchased from Charles River (Code 564). *il1rl1*^-/- mice were generously provided by Andrew McKenzie (Cambridge University). *il33*^-/- animals were obtained with a Materials Transfer Agreement from Amgen by Elia Tait Wojno, currently at the University of Washington and previously at Cornell University. All IL-33^-/- animals were housed at Cornell University. *il1rl1*-floxed embryos were received from KOMP repository (RRID MMRRC:048182-UCD). Importantly, we received notice after fully breeding these mice to cre-expressing mice, that the deposited floxed embryos contain a copy of an inducible, cardiomyocyte-driven cre [75], unknown to the KOMP repository. Upon receiving notice, we did detect the presence of MYH6cre^ERT2 alleles throughout our *il1rl1*^fl/fl colony. Importantly, we deleted *il1rl1* under the control of constitutive cre drivers and never administered tamoxifen to these mice. We performed statistical tests to determine the effect of MYH6cre expression, and did not find evidence of an effect in experiments where the MYH6cre was present in some animals and not others (S11 Fig). Moreover, in the experiments reported here (Fig 6), all animals were positive for the MYH6cre allele. All animals were housed in a UVA specific pathogen-free facility with a 12h light/dark cycle. Mice were age and sex matched for each experiment, and were sacrificed in parallel. Animals were infected with *T. gondii* at 7 to 9 weeks of age and were housed separately from breeding animals.

### Human brain tissue

Healthy human brain samples from adult patients were obtained from the UVA Human Biorepository and Tissue Research Facility. Samples were preserved on paraffin embedded slides. Patient identification and medical background was withheld and therefore IRB approval was not required.

## Parasite strains

The avirulent, type II *Toxoplasma gondii* strain Me49 was used for all infections (gift from Christopher Hunter, University of Pennsylvania). *T. gondii* cysts were maintained in chronically infected (1–6 months) Swiss Webster (Charles River) mice. To generate cysts for experimental infections, CBA/J (Jackson Laboratories) mice were infected with 10 cysts from brain homogenate of Swiss Webster mice by i.p. injection in 200µl PBS. 5–30 cysts from 4 week-infected CBA/J brain homogenate were then used to infect animals in all experiments.

## Immunohistochemistry

**Mouse tissue immunofluorescence.** Reporter mice were perfused with 30 mL PBS followed by 30 mL 4% PFA (Electron Microscopy Sciences). All non-reporter strains were only perfused with PBS. Brains were cut along the midline and post-fixed in 4% PFA for 24h at 4˚C. Brains were then cryoprotected in 30% sucrose (Sigma) for 24h at 4˚C, embedded in OCT (Tissue Tek), and frozen on dry ice. Samples were then stored at -20˚C. 40 µm sections were cut using a CM 1950 cryostat (Leica) and placed into a 24-well plate containing PBS. Sections were blocked in PBS containing 2% goat or donkey serum (Jackson ImmunoResearch), 0.1% triton, 0.05% Tween 20, and 1% BSA for 1h at RT. Sections were then incubated with primary antibody (anti-mouse IL-33 R&D systems Cat#AF3626 1:100, anti-human IL-33 R&D systems Cat#MAB36253 1:100, anti-mouse Olig2 EMD Millipore Cat#AB9610 1:1000, Anti-mouse GFAP Invitrogen Cat#13–0300 1:500, Anti-*T.gondii* was a gift from Fausto G. Araujo, Palo Alto Medical Foundation, Anti-mouse RFP Abcam Cat#ab62341 1:400, Anti-mouse iNOS Invitrogen Cat# PA5-16855 1:400, Anti-mouse mCherry Abcam Cat# ab167453 1:400, Anti-mouse Iba1 Abcam Cat#ab5076 1:500, Anti-mouse MHCII Invitrogen Cat#14-5321-85 1:500, Anti-mouse CC1 Millipore Sigma Cat#OP80 1:200, Anti-mouse CD3 ThermoFisher Cat#140032–82 1:500) diluted in blocking buffer at 4˚C overnight. Sections were washed the following day and incubated with secondary antibody (Donkey anti-goat AF488 Jackson Immunoresearch Cat#705-545-147 1:1000, Donkey anti-rat Rhodamine Jackson Immunoresearch Cat# 712-296-153 1:1000, Donkey anti-rabbit AF657 Jackson Immunoresearch Cat# 711-605-152 1:1000) in blocking buffer at room temperature for 1h. Sections were then washed and incubated with DAPI (Thermo Scientific) for 5 min at RT. Sections were then mounted onto Superfrost microscope slides (Fisherbrand) with aquamount (Lerner Laboratories) and coverslipped (Fisherbrand). Slides were stored at 4˚C before use. Images were captured using an TCS SP8 confocal microscope (Leica) and analyzed using Imaris (Bitplane) software. Volumetric analysis was achieved using the surfaces feature of Imaris.

**Human tissue immunofluorescence.** Slides containing 4 µm sections of human brain tissue were received from the UVA Biorepository and Tissue Research Facility and deparaffinized in a gradient from 100% xylene (Fisher) to 50% ethanol (Decon Laboratories). Slides were then washed in running water and distilled water. Antigen retrieval was performed by incubating slides in antigen retrieval buffer (10 mM sodium citrate, 0.05% Tween-20, pH 6.0) in an Aroma digital rice cooker for 45 min at 95˚C. Slides were then washed in running water followed by PBS-TW. Slides were then incubated with primary and secondary antibodies as described above for mouse brain tissue. Prior to imaging, Autoflourescence Eliminator Reagent was applied per the manufacturer's instructions (EMD Millipore Cat#2160).

**Propidium Iodide injection.** Adult naïve or 4-week infected mice were injected intraperitoneally with 20mg/kg propidium iodide (Invitrogen, cat#P1304MP). 24 hours post injection, the mice were sacrificed and their brains were fixed in 4% PFA and imaged for endogenous fluorescence by confocal microscopy.

**Tissue processing and flow cytometry.** Whole PBS-perfused brains were collected into 4 mL of complete RPMI (cRPMI)(10% fetal bovine serum, 1%NEAA, 1%Pen/Strep, 1%Sodium Pyruvate, 0.1%-β-mercaptoethanol). Papain digestion was performed for the chimera experiment. To perform papain digestion, brains were cut into 6 pieces and incubated in 5 mL HBSS containing 50U/mL DNase (Roche), and 4U/mL papain (Worthington-Biochem, Cat#LS003126) for 45 min at 37˚C. Tissue was triturated first with a large bore glass pipette tip, and twice with a small-bore pipette tip every 15 min. In all other experiments collagenase/dispase was used to digest brain tissue. To perform collagenase/dispase digestion, perfused brains were minced using a razor blade and passed through an 18-gauge needle. Brains were then digested with 0.227mg/mL collagenase/dispase (Sigma-Aldrich Cat#11097113001) and 50U/mL DNase1 (Millipore Sigma Cat#10104159001) for 1h at 37˚C. Following digestion, homogenate was strained through a 70 μm nylon filter (Corning, Cat#352350). Samples were then pelleted and spun in 20 mL 40% Percoll (GE Healthcare Cat#17-0891-09) at 650g for 25 min. Myelin was aspirated and cell pellets were washed with cRPMI. Finally, samples were resuspended in cRPMI and cells were enumerated. Meninges were collected from peeling from the brain and scoring from the skull cap and pooled. Meninges were passed through an 18-gauge needle five times and mashed through a 70μm filter. Samples were spun at 1500rpm for 10 min, and resuspended. Spleens were collected into 4 mL cRPMI and macerated through a 40 μM nylon filter (Corning, Cat#352340). Samples were pelleted and resuspended in 2 mL RBC lysis buffer (0.16 M NH$_4$Cl) Samples were then washed with cRPMI, and resuspended for counting and staining. In cases of acute infection, 4mL of peritoneal lavage fluid was pelleted and resuspended in 2mL of cRPMI for counting and staining. Single cell suspensions were pipetted into a 96 well plate and pelleted. Samples were resuspended in 50 μL Fc Block, made in FACS buffer (PBS, 0.2% BSA, and 2 mM EDTA) with 0.1 μg/ml 2.4G2 Ab (BioXCell, Cat#CUS-HB-197) and 0.1% rat gamma globulin (Jackson Immunoresearch, Cat#012-000-002), for 10 min. Cells were then surface stained in 50 μL FACS buffer for 30 min at 4˚C with directly conjugated antibodies (from eBioscience: Fixable viability dye eFluor 506 Cat#65-0866-18, CD3 FITC Cat#11-0031-85, CD62L FITC Cat#11-0621-85, CD8α PerCp-Cy5.5 Cat#45-0081-82, CD4 PE-Cyanine-7 Cat#25-0041-82, CD44 AF780 Cat#47-0441-82, MHCII FITC Cat#11-5321-82, CD45 PerCp-Cy5.5 Cat#45-0451-80, Ly6C PE-Cyanine-7 Cat#25-5932-82, CD11b AF780 Cat#47-0012-82, CD45.1 eFluor 450 Cat#48-0453-82, NK 1.1 FITC Cat#11-5941-85, CD19 FITC Cat#11-0193-82, ST2 PE Cat#12-9335-82, Thy1.2 (CD90) PB Cat#48-0902-80, FcεR1 APC Cat#17-5898-80; BD biosciences: CD45.2 FITC Cat#561874). Fixable viability dye was used at 1:800, all other antibodies were used at 1:200 dilution. Following surface staining, cells were fixed for at least 30 min at 4˚C with a fixation/permeabilization kit (eBioscience Cat#00-5123-43 and Cat#00-5223-56) and permeabilized (eBioscience Cat#00-8333-56). Samples were then incubated with intracellular antibodies in permeabilization buffer for 30 min at 4˚C (eBioscience: IFN-γ PerCp-Cy5.5 Cat#45-7311-82, IFN-γ APC Cat#17-7311-82, Ki67 APC Cat#17569880, iNOS APC Cat#17-5920-80, Foxp3 PB Cat#48-5773-82). For intracellular cytokine, samples were first incubated with PMA/ionomycin (Sigma-Aldrich Cat#P1585, Cat#I0634) and Brefeldin A (Selleckchem Cat#S7046) for 5 hours at 37˚C before surface staining. Samples were washed, resuspended in FACS buffer, and run on a Gallios flow cytometer (Beckman Coulter), and analyzed using Flowjo software, v. 10.

## qRT-PCR

Perfused brain tissue (100 mg) was placed into bead beating tubes (Sarstedt, Cat# 72.693.005) containing 1mL Trizol reagent (Ambion, Cat#15596018) and zirconia/silica beads (Biospec, Cat#11079110z). Tissue was homogenized for 30 seconds with a Mini-bead beater (Biospec) machine. RNA was extracted following homogenization per the Trizol Reagent manufacturer's

instructions. Complementary DNA was then synthesized using a High Capacity Reverse Transcription Kit (Applied Biosystems, Cat#4368813). Taqman gene expression assays were acquired from Applied Biosystems (*Ccl2* Cat#Mm00441242_m1, *Ccl5* Cat#Mm01302427_m1, *Cxcl9* Cat#Mm00434946_m1, *Cxcl10* Cat#Mm00445235_m1, *Cxcl1* Cat#Mm04207460_m1, *Vcam* Cat#Mm1320970_m1, *Icam* Cat#Mm00526023_m1, *Il33* Cat#Mm00505403_m1) or IDT (*Il1rl1*, S1 Table). A 2X Taq-based mastermix (Bioline, Cat#BIO-86005) was used for all reactions and run on a CFX384 Real-Time System (Bio-Rad). Hprt was used as the brain housekeeping gene (Applied Biosystems, Cat#Mm00446968_m1) and relative expression to wildtype controls was calculated as $2^{(-\Delta\Delta CT)}$.

For qRT-PCR assessment of parasite burden, brain tissue was placed in complete RPMI post-harvest, minced with a razor blade, and passed through an 18-gauge needle five times. Brain homogenate (300μL) was then put in a tube with zirconia/silica beads (Biospec, Cat#11079110z) and homogenized in a Mini-bead beater (Biospec) machine. Homogenate was then incubated at 65C for 3 hours with Lysis buffer from the Genomic DNA Isolate II Genomic DNA Kit (Bioline, Cat#BIO-52067). DNA was isolated from brain samples following the manufacturer's instructions. A 2X Taq-based mastermix (Bioline Cat#BIO-86005) was used for all reactions and run on a CFX384 Real-Time System (Bio-Rad), using *T. gondii* genomes as a standard curve, ranging in 10-fold dilutions from 300,000 genomes to 30 genomes. *T. gondii* genomes were isolated from cultured, infected human foreskin fibroblast cells. DNA from brain samples was diluted to 500ng/well. The Taqman assay used to detect *T. gondii* has been described previously [81].

### *T. gondii* cyst counts

Brain tissue (100 mg) was minced with a razor in 2mL cRPMI. Brain tissue was then passed through an 18-gauge and 22-gauge needle. 30 uL of resulting homogenate was pipetted onto a microscope slide (VWR) and counted on a Brightfield DM 2000 LED microscope (Leica). Cyst counts were extrapolated for whole brains.

### Bone marrow chimera

Wildtype B6.SJL-Ptprc[a] Pepc[b]/BoyJ (C57BL/6 CD45.1) and *il1rl1*[-/-] C57BL/6 CD45.2 mice were irradiated with 1000 rad. Irradiated mice received $3x10^6$ bone marrow cells from CD45.1 and CD45.2 donors the same day. Bone marrow was transferred by retro-orbital i.v. injection under isoflurane anesthetization. All mice received sulfa-antibiotic water for 2 weeks post-irradiation and were given 6 weeks for bone marrow to reconstitute. At 6 weeks, tail blood was collected from representative mice and assessed for reconstitution by flow cytometry. Mice were then infected for 4 weeks prior to analysis.

### Brain homogenate *ex vivo* supernatant collection

Brains were harvested from naïve and infected mice. Brain tissue was processed down to a single cell suspension as outlined in "tissue processing and flow cytometry". Cells from half a brain were incubated in a 96-well plate in 200μL for 4 hours at 37°C. Cells were then pelleted at 1500rpm for 5 min, and supernatant was stored at -80°C for measurement of IL-33 by ELISA.

### CSF collection

Naïve and infected mice were anesthetized with ketamine/xylazine, their necks were shaved and obstructing skin and tissue surrounding the dura of the cisterna magna was removed. A

small glass capillary was inserted into the dura of the cisterna magna, allowing approximately 10μL of CSF to fill the capillary. CSF from 4–5 mice was pooled and frozen at -80˚C.

### IL-33 ELISA

The IL-33 Quantikine ELISA kit was purchased from R&D Systems (Cat#M3300) and manufacturer instructions were followed. 50μL of either brain homogenate supernatant (*ex-vivo* assay) or CSF was used as sample volume. 4–5 CSF samples from individual animals were pooled to reach 50μL sample volume. IL-33 standards were serially diluted to a minimum of 30pg/mL. Final values of standards and samples were read on a spectrophotometer at 450nm.

### IFN-γ and IL-12 ELISAs

For ELISAs from day 7 post-infection, tail blood was taken from infected animals and incubated with 30U/mL heparin (Sigma, H3149-10KU) before being spun at 2000g for 10 min at 4˚C to obtain plasma. For the IFN-γ ELISA done at 12 days-post infection, blood was taken from cardiac puncture and allowed to sit for 30 min at room temperature before spinning at 2000g for 10 min at 4˚C to obtain serum. Quantikine ELISA kits for IFN-γ and IL-12p40 were purchased from R&D Systems (Cat# MIF00, Cat#M1240) and manufacturer instructions were followed, using 50μL of plasma or serum samples diluted twenty-fold. Final values of standards and samples were read on a spectrophotometer at 450nm.

### ACSA-2/CD11b magnetic enrichment

Brain tissue was harvested from mice and processed down to a single cell suspension as described in "tissue processing and flow cytometry" methods, except myelin removal beads (Miltenyi Cat#130096733) were used in place of Percoll. The Anti-ACSA2 and CD11b Microbead kits, purchased from Miltenyi (Cat#130097678, Cat#130093634), were used to enrich for astrocytes and macrophages, respectively, over magnetic columns (Miltenyi Cat#130-042-401). In the case of enriching for astrocytes, both kits were used to first remove CD11b+ cells and then enrich for ACSA-2+ cells. Double-purifications were performed in all cases, which was optional, but recommended, by Miltenyi's protocol. Cells were pelleted at 1500rpm and stored in 300μL Trizol (Ambion) at -80˚C until further use.

### Statistical analyses

Statistical analyses comparing two groups at one time point were done using a student's t-test in Prism software, v. 7.0a. Statistical analyses comparing more than two groups within the same timepoint or infection were done using a one-way Anova. In instances where data from multiple infections were combined, all from the same time-point post infection, a randomized block ANOVA was performed using R v. 3.4.4 statistical software to account for variability between infections. Genotype was modeled as a fixed effect and experimental day as a random effect. P values are indicated as follows: ns = not significant $p > .05$, * $p < .05$, ** $p < .01$, *** $p < .001$. The number of mice per group, test used, and p values are denoted in each figure legend. Data was graphed using Prism software, v.7.0a.

## Supporting information

**S1 Fig. IL-33 expression in mouse and human brain tissue. (A)** Real time PCR for *il33* transcript from whole brain homogente at 4 weeks post infection compared to naïve brain tissue. **(B)** Colocalization, denoted by gray arrows, of nuclear IL-33 protein (red) with mature oligodendrocytes, marked by nuclear Olig2 expression (green), and CC1(white) by confocal

fluorescence microscopy of infected mouse brain tissue. **(C and D)** Confocal fluorescence microscopy of nuclear IL-33 stain present in astrocytes (C) but not oligodendrocytes (D) in the temporal lobe of human brain tissue from patients that did not succumb to toxoplasmic encephalitis (healthy). Statistical significance was determined by randomized block ANOVA (A), which shows data pooled from two independent experiments * = p < .05, ** = p < .01, *** = p < .001. Scale bars indicate 50μm.
(TIF)

**S2 Fig. Characterization of inflammatory lesions in *T.gondii*-infected brain tissue. (A and B)** Representative images of immune cells surrounding foci of individual replicating parasites (green) in cortical brain tissue, including CD3+ T cells (red) (A), and MHCII+ (white) Iba1+ (red) myeloid cells (B). **(C and D)** Representative images of necrotic foci, featuring a loss of brain resident cells which express IL-33 (red), including GFAP+ astrocytes (green) (C), and Olig2+ oligodendrocytes (green) (D). **(E and F)** Representative images of propidium iodide fluorescence in brain tissue, 24 hours post i.p. injection into naïve and infected mice. Staining depicts propidium iodide (green), and GFAP+ astrocytes (red). Scale bars indicate 50μm in A-D, and 100μm or 30μm (insets) in E and F.
(TIF)

**S3 Fig. Brain parasite burdens at 12DPI (acute infection) versus 28DPI (chronic infection) in il1rl1-/- mice. (A and B)** Real time PCR for parasite genomic DNA from whole-brain homogenate of infected WT and il1rl1$^{-/-}$ mice at 12 days post infection (DPI) (A) and 28DPI (B). Statistical significance was determined by two tailed t-test (A) or a randomized block ANOVA (B), which shows data pooled from two independent experiments * = p < .05, ** = p < .01, *** = p < .001.
(TIF)

**S4 Fig. T cells in peripheral tissues and during acute infection in the absence of IL-33 signaling. (A)** Breakdown of total IFN-γ+ cells by cell type by flow cytometry in infected brain tissue four weeks post infection. IFN-γ was measured following stimulation *ex vivo* for five hours with PMA/ionomycin. **(B)** Assessment of spleen T cell numbers in *il1rl1*-deficient mice prior to infection by flow cytometry. **(C and D)** assessment of peripheral tissue T cell numbers and activation, including spleen (C) and blood (D) by flow cytometry 4 weeks post infection. **(E and F)** T cell numbers at day 10 acute infection by flow cytometry in the peritoneum (E) and blood (F). **(G-I)** plasma (G,I) or serum (H) ELISAs for IFN-γ (G,H) or IL-12 (I) during acute infection. Statistical significance was determined by randomized block ANOVA when two experiments were pooled (B,C, G, I), or by two-tailed t-test (D, E, F, H) * = p < .05, ** = p < .01, *** = p < .001.
(TIF)

**S5 Fig. The myeloid cell response is intact in peripheral tissues and during acute infection in the absence of IL-33 signaling. (A)** Assessment of spleen myeloid cell numbers in *il1rl1*-deficient mice prior to infection by flow cytometry. **(B and C)** assessment of peripheral tissue myeloid cell numbers and activation, including spleen (B) and blood (C) by flow cytometry 4 weeks post infection. **(D-F)** Myeloid numbers at day 10 during acute infection by flow cytometry in the peritoneum (D), spleen (E) and blood (F). Statistical significance was determined by randomized block ANOVA when two experiments were pooled (A and B), or by two-tailed t-test (C-F) * = p < .05, ** = p < .01, *** = p < .001.
(TIF)

**S6 Fig. Time course of il1rl1-/- brain immune cell populations during infection. (A-J)** Assessment of T cell and myeloid cell number and activation in the brain by flow cytometry at

12 days post infection (12DPI) (A-E) or 21 days post infection (21DPI) (F-J). Statistical significance was determined by two-tailed t-test (A-E), or by randomized block ANOVA when two experiments were pooled (F-J) * = p < .05, ** = p < .01, *** = p < .001.
(TIF)

**S7 Fig. Characterization of *ccl2* expression in naïve mice, and co-expression of *ccl2* and IL-33 by astrocytes. (A)** Real-time PCR analysis of whole-brain *ccl2* expression in naïve WT and *il1rl1*[-/-] mice. **(B)** Colocalization, denoted by white arrowheads, of nuclear IL-33 protein (red) with *ccl2* (green), both expressed by GFAP+ astrocytes (insets, white) by confocal fluorescence microscopy. **(C)** Quantification of frequency of colocalization of IL-33 and *ccl2* in cortical astrocytes. Statistical significance was determined by two-tailed t-test (A) * = p < .05, ** = p < .01, *** = p < .001. Scale bars indicate 30μm and 3μm(A, insets).
(TIF)

**S8 Fig. Magnetic enrichment for myeloid cells or astrocytes from infected brains. (A)** Unenriched single cell suspension of all purified cells from infected brain tissue 4 weeks post infection. **(B and C)** Assessment of purity achieved by enriching for myeloid cells using CD11b+ magnetic beads (B), or astrocytes (C), by negatively selecting for myeloid cells using CD11b+ magnetic beads, followed by positive selection for astrocytes with ACSA-2+ magnetic beads.
(TIF)

**S9 Fig. Characterization of ST2 expression on immune cells in the *T. gondii*-infected brain.** Detection of ST2 expression by flow cytometry of type 2 innate lymphoid cells, mast cells, regulatory T cells, effector T cells, and monocyte-derived macrophages in 4wk *T. gondii*-infected brain tissue.
(TIF)

**S10 Fig. Astrocyte *il1rl1* expression with infection, and astrocytic *il1rl1*-dependence for peripheral immune cell numbers. (A)** Cell-type specific magnetic enrichment for myeloid cells (CD11b+) or astrocytes (CD11b- and ACSA-2+) in naïve and chronically infected brain tissue. Single cell suspensions of enriched cells were resuspended in Trizol, RNA extracted, and measured by real time PCR for *ilrl1*(st2) expression. (B) Validation of excision of *il1rl1* from magnetically-enriched astrocytes in GFAPcre *il1rl1*[fl/fl] mice by quantitative PCR **(C)** Assessment of spleen immune cell numbers by flow cytometry four weeks post infection. Statistical significance was determined by one-way ANOVA with Tukey's test (A), a two-tailed t-test (B), or a randomized block ANOVA (C) * = p < .05, ** = p < .01, *** = p < .001.
(TIF)

**S11 Fig. Parasite burden in MYH6cre *il1rl1*[fl/fl] mice.** Parasite burden as measured by cyst count from brain homogenate. Statistical significance was determined by randomized block ANOVA.
(TIF)

**S1 Table. Custom primers for transmembrane il1rl1.** For specific detection of the IL-33 transmembrane receptor, rather than soluble ST2.
(XLSX)

## Acknowledgments

We thank Sachin P. Gadani and Kenneth S. Tung for the discussions regarding IL-33. We thank Marieke K. Jones for her guidance with statistical analysis and coding. We would like to

acknowledge the support we received from core facilities at the University of Virginia, including the Biorepository and Tissue Research Facility, the Flow Cytometry Core, and the Research Histology Core.

## Author Contributions

**Conceptualization:** Katherine M. Still, Nikolas W. Hayes, Tajie H. Harris.

**Formal analysis:** Katherine M. Still.

**Funding acquisition:** Elia D. Tait Wojno, Tajie H. Harris.

**Investigation:** Katherine M. Still, Samantha J. Batista, Carleigh A. O'Brien, Oyebola O. Oyesola, Simon P. Früh, Lauren M. Webb, Igor Smirnov, Michael A. Kovacs, Maureen N. Cowan, Nikolas W. Hayes, Jeremy A. Thompson, Elia D. Tait Wojno, Tajie H. Harris.

**Project administration:** Tajie H. Harris.

**Resources:** Elia D. Tait Wojno.

**Supervision:** Tajie H. Harris.

**Validation:** Katherine M. Still, Tajie H. Harris.

**Visualization:** Katherine M. Still.

**Writing – original draft:** Katherine M. Still.

**Writing – review & editing:** Katherine M. Still, Samantha J. Batista, Michael A. Kovacs, Tajie H. Harris.

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
