## [Decision Letter · Decision Letter 0]

21 Jun 2020

Dear Dr. Harris,

Thank you very much for submitting your manuscript "Astrocytes promote a protective immune response to brain Toxoplasma gondii infection via IL-33-ST2 signaling" for consideration at PLOS Pathogens. As with all papers reviewed by the journal, your manuscript was reviewed by members of the editorial board and by several independent reviewers. The reviewers appreciated the attention to an important topic. Based on the reviews, we are likely to accept this manuscript for publication, providing that you modify the manuscript according to the review recommendations.

All three reviewers felt that the manuscript was significant and warranted publication in PLoS Pathogens. However, each had a number of important comments and addressing them will improve the manuscript. Most significantly, further characterization of the astrocytic specificity of ST2 expression/function as well as staining for necrosis to further resolve IL-33 release mechanisms are issues that need to be resolved.. Two reviewers also requested that a more detailed time course be performed to further resolve mechanisms for how T-cells and monocytes are regulated by IL-33 signaling. These are indeed important issues and I recommend that you address them experimentally. If you choose not to, then a more detailed discussion about regarding the reviewers' points are required.

Sincerely,

Ira J Blader

Guest Editor

PLOS Pathogens

Kami Kim

Section Editor

PLOS Pathogens

Kasturi Haldar

Editor-in-Chief

PLOS Pathogens

orcid.org/0000-0001-5065-158X

Michael Malim

Editor-in-Chief

PLOS Pathogens

orcid.org/0000-0002-7699-2064

All three reviewers felt that the manuscript was significant and warranted publication in PLoS Pathogens. However, each had a number of important comments and addressing them will improve the manuscript. Most significantly, further characterization of the astrocytic specificity of ST2 expression/function as well as staining for necrosis to further resolve IL-33 release mechanisms are issues that need to be resolved.. Two reviewers also requested that a more detailed time course be performed to further resolve mechanisms for how T-cells and monocytes are regulated by IL-33 signaling. These are indeed important issues and I recommend that you address them experimentally. If you choose not to, then a more detailed discussion about regarding the reviewers' points are required.

Reviewer Comments (if any, and for reference):

Reviewer's Responses to Questions

**Part I - Summary**

Reviewer #1: In this study, Still and colleagues investigate the role of IL-33 during chronic Toxoplasma infection. The authors identify a key gap in our understanding that of the mechanism of innate sensing of the pathogen and subsequent immune cell recruitment to the brain. Using immunohistochemistry and direct measurement of IL-33 in CSF they show release of IL-33 during infection and identify cellular sources. Using mice deficient in the IL-33 receptor (ST2-/-) they demonstrate a requirement for IL-33 signaling for chemokine production, recruiting T cells and myeloid cells to the brain and the control of parasite burden. Finally using targeted cell specific deletion of ST2 and bone marrow chimeras, astrocytes are identified as the necessary source of IL-33 signaling. The data are rigorous and experiments are well done with appropriate conclusions. The ability to define astrocytes as a critical source of IL-33 responsiveness is not questioned. This is a significant contribution to our understanding of innate recognition mechanisms during infection in the CNS.

Reviewer #2: This is a very well-written paper with clearly presented data demonstrating that the IL33 receptor ST2 is important on astrocytes for control of chronic Toxoplasma burden and T cell numbers in the brain. The authors report that IL33 is expressed by oligodendrocytes and astrocytes, and released into the CSF by infection. They demonstrate that global knockout of either IL33 or it’s receptor ST2 leads to increased parasitic load and decreased immune responses during chronic infection. Finally, using bone marrow chimeras and constitutive cell-specific knockouts they claim that “Mice with IL33R-deficient astrocytes fail to promote an adaptive immune response and control infection.”

Reviewer #3: This important manuscript has revealed a new mechanism of host defense against the prevalent parasite, Toxoplasma gondii. Herein, the authors revealed a brain-specific mechanism of immunity mediated by IL-33.

Using a mouse model for chronic toxoplasmosis, the authors established the following key observations of IL-33-dependent immunity. The authors observed and clearly described that while IL-33 is constitutively expressed in the brain, infection can trigger production of this alarmin. Importantly, the authors reported that astrocytes are the major producers of this cytokine in the brain of infected mice. Through cleverly designed experiments utilizing Il1rl1 KO (ST2-deficient mice that lack a receptor for IL-33) mice, the authors concluded that IL-33 is required for the recruitment of IFN-g producing T cells and CCR2+ monocytes. Furthermore, the authors described that this recruitment was mediated by an indirect effect of IL-33 on astrocytes that were also responsible for producing CCL2. Finally, CCL2-mediated recruitment of monocytes to the brain parenchyma was also compromised in the absence of IL-33 signaling; however, this portion of the manuscript was not fully developed.

**Part II – Major Issues: Key Experiments Required for Acceptance**

Reviewer #1: 1. Stimulation of IL-33: Although a role for IL-33 in the CNS during disease has been established including during chronic Toxoplasma infection, the source and targets for this cytokine have not. The authors rightly point to the initial sensing of infection as a gap in our knowledge. IL-33 is presumably released following DAMP recognition however although referred to in the discussion evidence of necrosis has not been provided. Beautiful immunohistochemistry clearly shows the nuclear localization of IL-33 and the lack of it around areas of inflammation. Negative areas of IL-33 could be nuclear release or it could be changes in cellular composition in inflammatory foci. Staining for necrosis in these areas would help to establish the presence of IL-33-releasing DAMPs or suggest that necrosis is not the likely cause as also postulated in the discussion.

2. Source of IL-33: In Figure 1 strong evidence demonstrates oligodendrocytes as an important source of IL-33 while the measurement of IL-33 in the CSF is particularly impressive. GFAP and IL-33 overlap is not as convincing most likely due to the less than satisfying structural GFAP versus nuclear IL-33 discreet locations. Do in vitro cultured astrocytes produce IL-33? The authors provide an important piece of information about the high levels of nuclear IL-33 in baseline/uninfected brains. Does the cellular source of IL-33 change following infection? The relevance of this study is emphasized with data from human tissue demonstrating similar IL-33 nuclear localization. A minor point: the tissue is described as ‘healthy human’ – a rarity – more specifically is the human Toxoplasma seropositive or negative?

3. Acute versus chronic: In Figures 2 and 3 flow cytometry data demonstrate defects in T cell and monocyte recruitment. Such defects could be due to a failure to recruit activated and expanded peripheral cells or it could be a failure of activation and expansion in the periphery. Although this is acknowledged there could be additional data to underline the importance of CNS IL-33 in this peripheral immune cell recruitment. There are defects in T cell numbers in the blood and increases in T cells in the peritoneum at day 10 possibly suggesting a delay in recruitment/clearance. Perhaps serum IL-12/IFN-g could establish a normal kinetics? Or are T cells equivalently activated and expanded during acute infection? As this study is filling in gaps about initial recognition of infection in the brain (and in the absence of a recrudescent model) some data generated earlier during infection may be useful and could identify those initial sensing moments occurring in the CNS perhaps at the vasculature.

Reviewer #2: 1. Proof that astrocytes are the key cell type that expresses ST2 is lacking. The only analysis of cell-type specificity of ST2 is buried in supplementary fig 6, which is a cell-sorting experiment that demonstrates that gene expression is higher in CD11b+ cells in naïve mice and very low in astrocytes. Then post-infection, expression levels are reversed. There is a nice discussion of why this might be in the paper’s discussion. But I didn’t see any data on other cell types that might express ST2, or in situ or immunostaining to look at its cell-type specific expression during chronic Toxoplasma infection. This is also critical because Gfap-cre lines frequently have some or a lot of spill-over of expression into neurons and other cell types in the brain. Thus, some validation of the Gfap-cre, floxed ST2 mice to demonstrate the degree of knockout of the gene as well as the cell-specificity of ST2 expression during Toxoplasma infection would greatly strengthen the paper.

2. Is there any change in chemokine expression in the ST2 knockout mice prior to infection? These controls would help to determine if this is solely a defect in response to infection.

3. The data on the global knockouts of IL33 and ST2 demonstrate quite clearly that there is less inflammation and more parasites. However, there is no functional readout to say whether this is disadvantageous to the mice or not. One could imagine that actually less of an immune response is advantageous in a situation where the parasites do not grow further. Ideally, there would be some data on brain function and/or longitudinal effects on infectious burden, however if this is not possible due to lab shutdown, then the concept that either inflammation or parasite burden could be harmful should be discussed.

Reviewer #3: Major deficiencies:

1. The reliance on a single data point during T. gondii infection precludes deciphering the key cellular events regulated by IL-33. More specifically, it was not clear whether the deficiency in T cells resulted in reduced recruitment and activation of monocytes (measured by iNOS). Alternatively, impaired recruitment of CCR2+ monocytes may be responsible for the incomplete activation of T cells. This part of the manuscript was not fully developed and detailed kinetics would allow the authors to establish clearer associations between IL-33, T cells, and monocytes. This reviewer does not expect that the authors will deplete the cells in question as this would fatally compromise host defense against the parasite; instead, a detailed kinetic analysis of monocyte and T cell recruitment mediated by IL-33 would fully address this question.

2. The data suggest that astrocytes are the producers of both IL-33 and CCL2 as the later mediator was triggered in an ST2-dependent manner, a signaling event seen in astrocytes; however, further examining whether astrocytes are indeed co-producers of these factors would significantly bolster the author’s findings. This reviewer feels the authors have already developed the essential experimental tools to determine if astrocytes are indeed co-producers of these soluble mediators.

**Part III – Minor Issues: Editorial and Data Presentation Modifications**

Reviewer #1: the data are well presented

Reviewer #2: 1. Cell-specificity of IL33. The data shows that oligodendrocytes have at least as much IL33 expression as astrocytes, and likely more in the white matter. This should be acknowledged, ie that they both make it but are signaling through astrocytes alone. The role of oligodendrocytes is presented in the paper but is not reflected in the title and other places in the text.

2. Similarly, the last line of the abstract implies that without astrocytic ST2 there is no adaptive immune response. This seems like an overstatement, the data appear to show that there is a reduced, inadequate response.

3. In lines 309-310, the authors claim IL-33-ST2 signaling on astrocytes affects infiltrating myeloid cell numbers but Fig 6D doesn’t show that. This should be edited, and also highlighted in the discussion.

4. In the abstract, the authors state that “IL-33 signaling promotes chemokine expression within brain tissue and is required for the recruitment of peripheral anti parasitic immune cells, including IFN-g-expressing T cells and iNOS-expressing monocytes.” However they also show deficits in T cell proliferation, so it is not clear whether the decreased number of T cells is due to decreased recruitment, proliferation, or both. This sentence should be rephrased, and it would be appropriate to mention that it might be either mechanism in the discussion.

5. Fig 5 is a little confusing. The diagram at the top could be utilized to make the labels in the graphs below more clear. Specifically it’s hard to figure out which is the BM and which is the host. Also the methods make it clear that SJL mice were not used but rather a congenic B6 strain carrying the Ptprc allele of SJL. SJL mice, and actually even sometimes the congenics, can have different immune responses.

6. Please list the antibodies used for immunohistochemistry, their sources, and their dilutions. Also, please clarify - was the IL33 actually detected by in situ or was it detected by immunohistochemistry in Figs 1 and S1? It looks so nuclear and the figure legends refer to “IL33 expression” but I didn’t see this in the methods.

7. For Fig. 1B the authors should state their Ns for quantification in the figure legend.

8. Line 156 needs the reference “Figure 1E.”

9. Given that there is no iNOS expression in the periphery in both WT and ST2-/- infected mice, is there any expression in naïve mice? Or if iNOS expression has been observed in the periphery in other infection models, it could be worth mentioning to draw distinctions from Toxoplasma gondii’s mechanism of infection.

10. Fig 5E-G might be more clear if only the significant relationships are annotated in the bar graphs, and if there were more space over the graph to separate the multiple comparisons.

11. In Fig 6B and 6E, it’s interesting that microglia and astrocytes have reverse trends for CD4+ cell numbers. In the Cx3cr1-Cre constitutive KO, this is the only phenotype that is affected, whereas in the Gfap-Cre constitutive KO, it is one of the only phenotypes that isn’t affected. This makes sense since there is a connection between CD4+ T cells and MHC II expressing microglia, so perhaps it could be worth commenting on.

12. Line 356 missing citation for in vitro experiment.

13. Line 368 missing citation for microglia.

Reviewer #3: Minor comments:

1. ST2 KO mice should be abbreviated as Il1rl1 KO

2. The significance of the manuscript will be improved if the authors can establish a mechanism of IL-33 release in response to T. gondii infection. It was not clear whether the infected cells or neighboring cells release this alarmin during the parasitic infection.

PLOS authors have the option to publish the peer review history of their article (what does this mean?). If published, this will include your full peer review and any attached files.

Reviewer #1: No

Reviewer #2: Yes: Marion Buckwalter

Reviewer #3: Yes: Felix Yarovinsky
---

## [Decision Letter · Decision Letter 1]

29 Sep 2020

Dear Dr. Harris,

We are pleased to inform you that your manuscript 'Astrocytes promote a protective immune response to brain Toxoplasma gondii infection via IL-33-ST2 signaling' has been provisionally accepted for publication in PLOS Pathogens.

Best regards,

Ira J Blader

Guest Editor

PLOS Pathogens

Kami Kim

Section Editor

PLOS Pathogens

Kasturi Haldar

Editor-in-Chief

PLOS Pathogens

orcid.org/0000-0001-5065-158X

Michael Malim

Editor-in-Chief

PLOS Pathogens

orcid.org/0000-0002-7699-2064

Reviewer Comments (if any, and for reference):

Reviewer's Responses to Questions

**Part I - Summary**

Reviewer #1: Still and colleagues present a novel body of work that demonstrates innate sensing of the pathogen Toxoplasma in the brain. Comprehensive experiments using IL-33R-deficient mice show the role of this cytokine in signaling to CNS resident cells - likely astrocytes to enable protective immunity specifically in the brain. The revised version of this fully addresses the reviewers comments with new experiments and data including kinetic and cell death analysis while acknowledging some limitations in the discussion and text. This work is useful to the Toxoplasma community but will also provide important information on innate recognition and mechanisms of immune defense in the brain.

Reviewer #2: Thank you for your thoughtful responses to our concerns, we appreciate them and continue to feel that this is an important and well-written paper with solid data.

Reviewer #3: All my comments have been fully addressed

**Part II – Major Issues: Key Experiments Required for Acceptance**

Reviewer #1: none

Reviewer #2: Our only remaining concern is the fact that oligodendrocytes as well as astrocytes produce IL33 and that although the data is convincing that astrocytes have IL1rl1/ST2 and play a role in inflammation, it is possible that other cells are responding to IL33 as well.

The authors note this on line 462 in the discussion: "Notably, immune defects in GFAPcre il1rl1fl/fl mice did not match the magnitude observed in il1rl1-/- mice, and did not significantly impact myeloid cell recruitment. These results either implicate compensation by other cell types when il1rl1 is deleted from a single population, or the presence of additional il1rl1-expressing radio-resistant cell types not considered here, such as neurons, oligodendrocytes, and endothelial cells, which have recently been shown to express il1rl1 during disease and can be directly activated in response to IL-33 in vitro (69, 80)."

In the results section, we believe they should state clearly that the astrocyte knockout was incomplete, and put the data that validates their mouse model in the main paper and not a supplemental figure. In addition, we think they should state more clearly in the discussion that their data, while demonstrating a clear and important role of astrocytes, does not rule out that other brain cell types may contribute to the effects of IL33/Il1rl1 signaling.

Reviewer #3: (No Response)

**Part III – Minor Issues: Editorial and Data Presentation Modifications**

Reviewer #1: (No Response)

Reviewer #2: (No Response)

Reviewer #3: (No Response)

PLOS authors have the option to publish the peer review history of their article (what does this mean?). If published, this will include your full peer review and any attached files.

Reviewer #1: No

Reviewer #2: No

Reviewer #3: **Yes: **Felix Yarovinsky

---

## [Editor Report · Acceptance letter]

19 Oct 2020

Dear Dr. Harris,

We are delighted to inform you that your manuscript, "Astrocytes promote a protective immune response to brain Toxoplasma gondii infection via IL-33-ST2 signaling," has been formally accepted for publication in PLOS Pathogens.

Best regards,

Kasturi Haldar

Editor-in-Chief

PLOS Pathogens

orcid.org/0000-0001-5065-158X

Michael Malim

Editor-in-Chief

PLOS Pathogens

orcid.org/0000-0002-7699-2064